# Between Linear and Sinusoidal:
# Rethinking the Time Encoder in Dynamic Graph Learning

**Hsing-Huan Chung**                                                    *hhchung@utexas.edu*
*Department of Electrical and Computer Engineering*
*University of Texas at Austin*

**Shravan Chaudhari**                                                    *schaud35@jh.edu*
*Department of Computer Science*
*Johns Hopkins University*

**Xing Han**                                                    *xhan56@jhu.edu*
*Department of Computer Science*
*Johns Hopkins University*

**Yoav Wald**                                                    *yoav.wald@nyu.edu*
*Center for Data Science*
*New York University*

**Suchi Saria**                                                    *ssaria@cs.jhu.edu*
*Department of Computer Science*
*Johns Hopkins University*

**Joydeep Ghosh**                                                    *jghosh@utexas.edu*
*Department of Electrical and Computer Engineering*
*University of Texas at Austin*

**Reviewed on OpenReview:** *https://openreview.net/forum?id=W6GQvdOGHg*

## Abstract

Dynamic graph learning is essential for applications involving temporal networks and requires effective modeling of temporal relationships. Seminal attention-based models like TGAT and DyGFormer rely on sinusoidal time encoders to capture temporal dependencies between edge events. Prior work justified sinusoidal encodings because their inner products depend on the time spans between events, which are crucial features for modeling inter-event relations. However, sinusoidal encodings inherently lose temporal information due to their many-to-one nature and therefore require high dimensions. In this paper, we rigorously study a simpler alternative: the linear time encoder, which avoids temporal information loss caused by sinusoidal functions and reduces the need for high-dimensional time encoders. We show that the self-attention mechanism can effectively learn to compute time spans between events from linear time encodings and extract relevant temporal patterns. Through extensive experiments on six dynamic graph datasets, we demonstrate that the linear time encoder improves the performance of TGAT and DyGFormer in most cases. Moreover, the linear time encoder can lead to significant savings in model parameters with minimal performance loss. For example, compared to a 100-dimensional sinusoidal time encoder, TGAT with a 2-dimensional linear time encoder saves **43%** of parameters and achieves higher average precision on five datasets. While both encoders can be used simultaneously, our study highlights the often-overlooked advantages of linear time features in modern dynamic graph models. These findings can positively impact the design choices of various dynamic graph learning architectures and eventually benefit temporal network applications such as recom-

mender systems, communication networks, and traffic forecasting. The experimental code is available at: https://github.com/hsinghuan/dg-linear-time.git.

# 1 Introduction

Dynamic graph learning (Longa et al., 2023; Yu et al., 2023; Huang et al., 2024) has gained significant attention in recent years due to its wide range of applications, including recommender systems (Zhang et al., 2022), traffic forecasting (Yu et al., 2017), and anomaly detection (Wang et al., 2021a). A key task in dynamic graph learning is future link prediction (Poursafaei et al., 2022; Qin & Yeung, 2023), where the goal is to predict future connections or interactions between nodes in a temporal graph based on historical data. Temporal features such as the interaction time play an important role in capturing the dynamic nature of these graphs. For instance, in a temporal person-to-person communication network, imagine an individual who exchanged several emails with Alice this morning but last contacted Bob weeks ago. The next message is far more likely to go to Alice than to Bob, underscoring how the timing of interactions must be captured by dynamic graph models when predicting future links.

One popular approach for encoding temporal information in dynamic graph learning is the sinusoidal time encoder. Initially introduced in the event sequence modeling literature (Xu et al., 2019) to make time encodings compatible with the self-attention mechanism, the sinusoidal time encoder maps timestamps or time differences as periodic functions of their values, represented as multi-dimensional sinusoidal time encodings. Temporal Graph Attention (TGAT) (da Xu et al., 2020) introduced the sinusoidal time encoder to dynamic graph learning, and subsequent works (Wang et al., 2021c; Cong et al., 2023; Yu et al., 2023) have widely adopted it as the default approach. Motivated by the importance of temporal information in dynamic graphs, we revisit the choice of time encoders in this study.

We examine a simpler alternative: the linear time encoder. Intuitively, sinusoidal functions are many-to-one mappings, which lead to inevitable temporal information loss, while the linear time encoder avoids it by representing time through one-to-one mappings. Previous work (Xu et al., 2019; da Xu et al., 2020) motivated sinusoidal encodings by showing that their inner products depend on the time spans between events, which are essential features for modeling time-sensitive relationships. However, we prove that self-attention can also compute time spans effectively from linear time encodings. We demonstrate this through experiments on a synthetic task where such features are required for prediction. These observations motivate us to evaluate the linear time encoder in dynamic graph learning. We conduct extensive experiments on future link prediction tasks across six dynamic graph datasets. Our results reveal that, out of 24 model-dataset combinations, the linear time encoder outperforms sinusoidal time encoder variants in 19 and 18 cases under random and historical negative sampling, respectively. The largest performance gains caused by switching from the sinusoidal to linear time encoder on TGAT and DyGFormer (Yu et al., 2023) are 22.48 and 7.28 increases in average precision (AP) scores under historical negative sampling, respectively. Additionally, we show that we can reduce the dimensionality of linear time encodings in TGAT and save 43% of model parameters, with only a maximum decrease of 2.44 in AP. In contrast, sinusoidal time encoding suffers a maximum drop of 8.08 in AP under the same model size reduction. We perform similar experiments on reducing the temporal feature channel dimension in DyGFormer (Yu et al., 2023). While both sinusoidal and linear time encoders experience similar performance drops on two datasets, the performance of linear time encoders surprisingly improves on two other datasets, with 38% and 34% savings in model parameters, whereas sinusoidal time encoders continue to experience a performance decline. These results confirm that linear time encoders are not only better performing but also more cost-effective than the widely used sinusoidal time encoders.

In summary, our contribution is threefold. First, we examine the linear time encoder as an alternative to the widely used sinusoidal time encoder, offering a simpler and more efficient method for encoding temporal information in dynamic graph learning. Second, we prove that self-attention can compute time spans between events from linear time encodings and demonstrate it can learn such capabilities through experiments on synthetic datasets. Finally, we present extensive experimental results across six dynamic graph datasets, showing that the linear time encoder outperforms sinusoidal variants in most cases, while also being more cost-effective by reducing model parameters without significantly compromising performance.

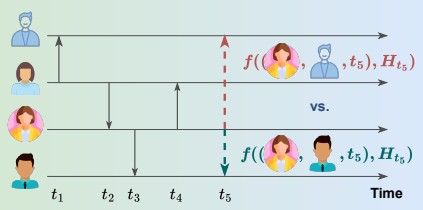

Figure 1: Future link prediction on a continuous-time dynamic graph. At time $t_5$, the link predictor $f$ ranks how likely edges will form given the historical edge events happening at $t_1, \ldots, t_4$.

## 2 Preliminaries

We revisit the future link prediction problem on continuous-time dynamic graphs and review two seminal attention-based models: Temporal Graph Attention (TGAT) (da Xu et al., 2020) and DyGFormer (Yu et al., 2023). Then, we describe the widely-used sinusoidal time encoder. We focus on attention-based models because self-attention is prevalent in mainstream neural architectures for not only text (Waswani et al., 2017) and vision (Dosovitskiy et al., 2021) but also graphs (Veličković et al., 2018; Dwivedi & Bresson, 2021; Kim et al., 2022). We investigate the two models in specific because TGAT introduced the sinusoidal time encoder to work with graph attention network without additional heuristics and DyGFormer is the state-of-the-art transformer-based dynamic graph model. In Section 3, we motivate the use of linear encodings by proving that self-attention can compute time spans between events from these encodings, making this applicable to both models. Additionally, as TGAT and DyGFormer are representative graph and sequence-based models, our results apply to both paradigms.

### 2.1 Future Link Prediction

Future link prediction is the problem of predicting whether an edge will appear in a *continuous-time dynamic graph*.

**Definition 2.1** (*Continuous-Time Dynamic Graph*)**.** A continuous-time dynamic graph over a time interval $\mathcal{I}_T := [0, T]$ is an ordered pair $\mathcal{G} = (\mathcal{V}, \mathcal{E})$ where $\mathcal{V} = \{1, \ldots, N\}$ is a set of nodes and $\mathcal{E} \subseteq \{(i, j, t) \in \mathcal{V}^2 \times \mathcal{I}_T\}$ is a set of edges representing the interaction events between nodes where $i$ is the source node, $j$ is the destination node, and $t$ is the time of interaction. Depending on individual datasets, each node $i \in \mathcal{V}$ may be associated with node attributes $\mathbf{x}_i \in \mathbb{R}^{d_V}$ and each edge $(i, j, t) \in \mathcal{E}$ may also be associated with edge attributes $\mathbf{x}_{i,j}^t \in \mathbb{R}^{d_E}$.

Future link prediction (Poursafaei et al., 2022; Qin & Yeung, 2023) involves assigning a score that estimates how likely it is that a target edge $(i, j, t)$ will form at time $t$ given the history of interactions up to that point. We call an edge $(i, j, t)$ positive if $(i, j, t) \in \mathcal{E}$ and negative otherwise. Evaluation for future link prediction typically treats the task as a binary classification problem between positive and negative edges, using ranking metrics such as Area Under the Curve (AUC) or Average Precision (AP). We expand on different ways of sampling negative edges in Section 4.1.2.

### 2.2 Attention-Based Dynamic Graph Learning Models

To estimate how likely a target edge $(i, j, t)$ is to occur, a typical dynamic graph learning model computes the temporal representations of node $i$ and $j$ according to their first-order or higher-order historical edge interactions. Using node $i$ as an example, the first-order historical edge interactions involving node $i$ are denoted as $\mathcal{S}_i^t := \{(i, v, t') | t' < t, (i, v, t') \in \mathcal{E}\} \cup \{(u, i, t') | t' < t, (u, i, t') \in \mathcal{E}\}$. The first-order temporal neighbors of node $i$ are denoted as $\mathcal{N}_i^t := \{(v, t') | (i, v, t') \in \mathcal{S}_i^t \vee (v, i, t') \in \mathcal{S}_i^t\}$. The temporal representations of node $i$ and $j$ are later concatenated and forwarded through a multi-layer perceptron (MLP) to obtain a final score on how likely $(i, j, t)$ will happen. Below, we briefly introduce two representative attention-

based dynamic graph learning models, TGAT and DyGFormer. For a more detailed review, please refer to Appendix A.

TGAT is a graph-based model that recursively aggregates higher-order temporal neighbor information through multiple layers. The initial temporal representation of node $i$, $\tilde{\mathbf{h}}_i^{(0)}(t)$, is its raw attributes $\mathbf{x}_i$. To compute the $l$'th layer temporal representation of node $i$, $\tilde{\mathbf{h}}_i^{(l)}(t)$, from the $l-1$'th layer, TGAT uses self-attention where node $i$ acts as the query and its temporal neighbors act as the keys and values. Let $\Phi : \mathcal{I}_T \to \mathbb{R}^{d_T}$ denote a time encoder that maps time values to $d_T$-dimensional time encodings. For each node $v$ and interaction time $t'$ involved, the input features to self-attention consists of the $l-1$'th layer temporal node representation $\tilde{\mathbf{h}}_v^{(l-1)}(t')$ and a time encoding that encodes the difference between $t'$ and the target time $t$, i.e. $\Phi(t-t')$. After aggregating information from temporal neighbors into a vector, it is concatenated with $\mathbf{x}_i$ and forwarded through an MLP to get $\tilde{\mathbf{h}}_i^{(l)}(t)$.

DyGFormer is a sequence-based model that considers the first-order temporal neighbors of a node when computing its representation. To compute the temporal representation of node $i$ at time $t$, the target edge and the historical edge interactions involving $i$, $\mathcal{S}_i^t$, are retrieved and sorted by the timestamps in ascending order. The features corresponding to each edge are collected, including neighbor node attributes, edge attributes, neighbor co-occurrence features, and time encodings $\Phi(t - t')$ where $t'$ is the edge timestamp. The target edge attributes are filled with a zero vector since the target edge has not occurred. Eventually, the four types of features are each linearly transformed into a channel dimension of $d_{\text{ch}}$ and concatenated together as the input sequence matrix to the transformer $\mathbf{X}_i^t$. Later, the sequence matrices $\mathbf{X}_i^t$ and $\mathbf{X}_j^t$ corresponding to the two nodes of the target edge $(i, j, t)$ are concatenated in the sequence dimension and forwarded through an $L$-layer transformer to output $\mathbf{Z}^{(L)}(t)$, which contains edge interaction representations of both nodes. Finally, DyGFormer applies mean pooling and a linear transformation to the positions corresponding to $i$ and $j$ in $\mathbf{Z}^{(L)}(t)$ to get their respective temporal representations.

### 2.3 Sinusoidal Time Encoder

Time encodings are essential for capturing the temporal relationships between events. Importantly, the time spans between events matter more than their absolute timestamps. For example, the influence of an edge interaction at time $t_1$ to another one at $t_2$ should depend on $t_1 - t_2$. Motivated by this intuition, Xu et al. (2019) propose a time encoder that ensures the inner product between encodings reflects the time span. They achieve this with the sinusoidal encoder $\Phi(t) = \sqrt{\frac{1}{d_T}}[\cos(\omega_1 t) \sin(\omega_1 t) \ldots \cos(\omega_{d_T} t) \sin(\omega_{d_T} t)]^\top$ with learnable parameters $\omega_1, \ldots, \omega_{d_T} \in \mathbb{R}$, which satisfies the desired property: $\langle \Phi(t_1), \Phi(t_2) \rangle = \frac{1}{d_T} \sum_{i=1}^{d_T} \cos(\omega_i (t_1 - t_2))$. The sinusoidal time encoder is then introduced to dynamic graph learning in TGAT (da Xu et al., 2020). Several subsequent models such as GraphMixer (Cong et al., 2023) and DyGFormer (Yu et al., 2023) follow this choice. Their implementations use all cosine functions with phases to encode the difference between the target time $t$ and an edge event time $t'$ for the edge: $\Phi(t-t') = [\cos(\omega_1(t-t')+\phi_1) \ldots \cos(\omega_{d_T}(t-t')+\phi_{d_T})]^\top$. This is a valid design choice since cosines are as expressive as sines, and since only the time span between events matters, the time reference point can be shifted without affecting the span: $|t_1-t_2| = |(t-t_1)-(t-t_2)|$. While GraphMixer uses fixed frequency and phase parameters, TGAT and DyGFormer make them learnable, so we focus on learnable time encoders in this study.

## 3 Do We Need Sinusoidal Time Encodings?

### 3.1 Linear Time Encoder

Although the inner products of sinusoidal encodings can be expressed as a function of the time differences, both the encodings and the inner products are many-to-one mappings from time values to a fixed range $[-1, 1]$, which inevitably loses information. We consider a simple linear time encoder $\Phi : \mathcal{I}_T \to \mathbb{R}^{d_T}$, to encode the difference between the target time $t$ and an edge event time $t'$, $\Delta t' := t - t'$:

$$\Phi(\Delta t') = \begin{bmatrix} w_1 \Delta t' + b_1 & \ldots & w_{d_T} \Delta t' + b_{d_T} \end{bmatrix}^\top \tag{1}$$

where $w_1, \ldots, w_{d_T} \in \mathbb{R}$ and $b_1, \ldots, b_{d_T} \in \mathbb{R}$ are learnable weights and biases parameters. In practice, we encode the standardized time difference value $(\Delta t' - \mu_{\Delta t'})/\sigma_{\Delta t'}$ where $\mu_{\Delta t'}$ and $\sigma_{\Delta t'}$ are the empirical mean and standard deviation of observed time differences for numerical stability and optimization efficiency purposes. Encoding standardized time differences will still result in time encodings linear to the original time differences. Unlike sinusoidal time encodings, each dimension $i$ of a linear time encoding is a one-to-one mapping from $\Delta t'$ as long as $w_i \neq 0$. On the other hand, linear time encodings do not satisfy the inner product property that the sinusoidal time encodings do: $\langle \Phi(t_1), \Phi(t_2) \rangle = \psi(t_1 - t_2)$. Nevertheless, we show that self-attention can compute the time spans between events from linear time encodings in the following proposition.

**Proposition 3.1.** *Let $\boldsymbol{x} = \{(\mathbf{x}_1, t_1), \ldots, (\mathbf{x}_M, t_M)\}$ be a set of events where each event $m$ contains a feature vector $\mathbf{x}_m \in \mathbb{R}^d$ and a timestamp $t_m \in \mathcal{I}_T$. Let $t$ be the target time. There exists a linear time encoder $\Phi : \mathcal{I}_T \to \mathbb{R}^{d_T}$, a query weight matrix $\mathbf{W}_Q \in \mathbb{R}^{(d_T+d) \times d_h}$, and a key weight matrix $\mathbf{W}_K \in \mathbb{R}^{(d_T+d) \times d_h}$ that can factor the time span $t_m - t_n$ between any two events $m, n$ into the attention score:*

$$\mathbf{q}_m = \mathbf{W}_Q^\top \begin{bmatrix} \Phi(t - t_m) \\ \mathbf{x}_m \end{bmatrix}, \ \mathbf{k}_n = \mathbf{W}_K^\top \begin{bmatrix} \Phi(t - t_n) \\ \mathbf{x}_n \end{bmatrix}$$

$$\langle \mathbf{q}_m, \mathbf{k}_n \rangle = h(t_m - t_n) + g(\mathbf{q}_m, \mathbf{k}_n)$$

*where $h : \mathbb{R} \to \mathbb{R}$ models the patterns relevant to the time span and $g : \mathbb{R}^{d_T+d} \times \mathbb{R}^{d_T+d} \to \mathbb{R}$ models other patterns between the events.*

*Proof.* Our proof provides a construction of $\Phi$, $\mathbf{W}_Q$, and $\mathbf{W}_K$ that only requires specifying two dimensions of the parameters. Please refer to Appendix B for the complete proof. □

Proposition 3.1 shows that there exists a particular combination of a linear time encoder and self-attention weights that computes the time spans. To understand whether such a combination can be learned from data, we design a simple but representative synthetic task of event sequence classification and test whether a linear time encoder and a transformer can successfully learn the task.

### 3.2 Synthetic Task: Event Sequence Classification

#### 3.2.1 Data-Generating Process

Consider sequence-label pairs with the form of $(\{(x_1, t_1), \ldots, (x_M, t_M), t\}, y)$ where $x_m$ and $t_m$ are the feature value and timestamp of event $m$, $t$ is the target time, and $y$ is the sequence label. The data-generating process is as follows:

$$x_m = \begin{cases} 1, & \text{with probability } 0.5 \\ -1, & \text{otherwise} \end{cases}$$

$$t_1 \sim \text{Exp}(\lambda)$$

$$\Delta t_m \sim \text{Exp}(\lambda), \ t_m = t_{m-1} + \Delta t_m \text{ for } m = 2, \ldots, M$$

$$\Delta t \sim \text{Exp}(\lambda), \ t = t_M + \Delta t$$

$$\epsilon \sim \mathcal{N}(0, 0.01)$$

$$y = \mathbb{1}[\sum_{m=1}^{M} \exp(-\alpha(t - t_m))x_m + \epsilon > 0] \tag{2}$$

where the intensity rate $\lambda$ and exponential decay rate $\alpha$ are the parameters. The labeling function resonates with the assumption that the more recent a historical edge event takes time, the more influence it has on the current state of a node.

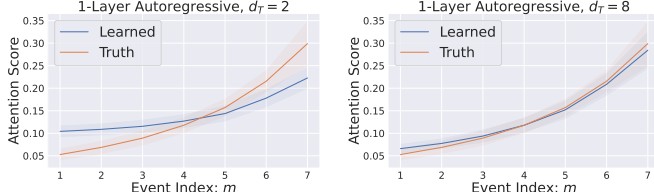

Figure 2: Test accuracies of transformers vs. $d_T$ on the synthetic task. The left two figures are the results of full self-attention and autoregressive attention of 1-layer transformers. The right two are the results of 2-layer transformers.

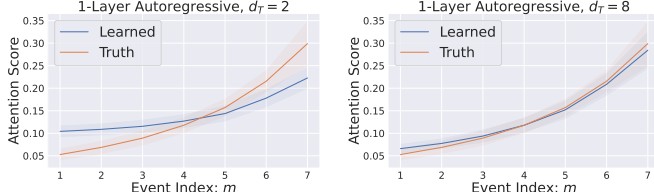

Figure 3: Average attention scores for each event index $m$ of 1-layer autoregressive transformers with linear time encodings. We compare the attention scores when using $d_T = 2$ and $8$ with the true attention scores derived from the labeling function in Eq. 2.

### 3.2.2 Event Sequence Classification With Transformers

We encode the event sequence with an $L$-layer transformer into a sequence representation and classify it with a linear head. The input $\mathbf{X} \in \mathbb{R}^{(M+1) \times (d_T + 1)}$ is organized as:

$$
\mathbf{X} = \begin{bmatrix} \Phi(t - t_1)^\top & x_1 \\ \vdots & \vdots \\ \Phi(t - t_M)^\top & x_M \\ \Phi(0)^\top & 0 \end{bmatrix}
$$

We follow the transformer layer implementation in DyGFormer as reviewed in Appendix A. Let $d_h$ denote the attention dimension and $\mathbf{Z}^{(l)} \in \mathbb{R}^{(M+1) \times d_h}$ denote the $l$'th layer sequence representation. We experiment with full self-attention and autoregressive attention. In full self-attention, we follow DyGFormer by averaging $\mathbf{Z}^{(L)}$ over the sequence dimension to obtain the sequence representation: $\mathbf{z} = \frac{1}{M+1} \mathbf{Z}^{(L)^\top} \mathbf{1}_{M+1}$. For autoregressive attention, we apply a causal mask that prevents an event from aggregating information about any future events. Following the convention of autoregressive transformers (Radford et al., 2018), we take the last "token" representation as the sequence representation: $\mathbf{z} = \mathbf{Z}^{(L)^\top} \mathbf{e}_{M+1}$. After getting the sequence representation, we apply a linear head for binary classification. We train the transformer and linear head by minimizing the binary cross entropy loss between the prediction and $y$.

### 3.2.3 Experiments

We conduct experiments to see if transformers with linear time encoders can learn the event sequence classification task. We use $\lambda = 0.01$, $\alpha = 0.003$, and $M = 7$ to generate 2000 sequences in total where 70%/15%/15% of the sequences are for train/validation/test. We experiment with 1-layer and 2-layer transformers and vary $d_T$ in between $\{2, 4, 8, 16\}$. We compare linear time encodings, sinusoidal time encodings with all cosines, and sinusoidal time encodings with sine and cosine pairs. We do 10 runs for each configuration and present the results in Figure 2.

The results show that transformers with linear time encoders are capable of learning the event sequence classification task even when $d_T$ is small. In Figure 3, we also visualize the attention scores of 1-layer autoregressive transformers with linear time encodings, which essentially uses $(\mathbf{X})_{M+1,:}$ as the query and $(\mathbf{X})_{1:M,:}$ as the keys and values. We know the true attention scores based on the labeling function in Eq.

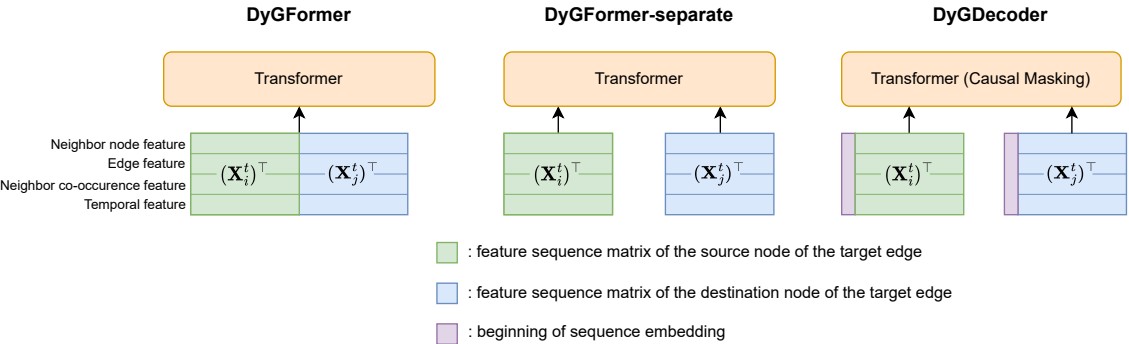

Figure 4: An illustration of the DyGFormer variants that we experimented with on dynamic link prediction. DyGFormer-separate and DyGDecoder forward the source and destination node feature sequences separately through the transformers while DyGFormer concatenates them and forwards jointly. In addition, DyGDecoder applies causal masking in its self-attention blocks and prepends a learnable "beginning of sequence" embedding to the input feature sequence matrices.

2. We average the learned and true attention scores over all sequences per event index and compare the two in the plots. According to the plots, the 1-layer transformer can learn the exponential decaying pattern when $d_T = 2$ and becomes closer to the true labeling function when $d_T = 8$. Our experiments confirm that self-attention can learn to compute time spans from linear time encodings and extract relevant patterns. Motivated by this result, we extend our empirical study to future link prediction on dynamic graphs.

## 3.3 Modifying TGAT and DyGFormer

We study how different time encoders perform with TGAT and DyGFormer in the context of future link prediction. For TGAT, we replace its sinusoidal time encoder with a linear time encoder and compare their performances. For DyGFormer, we consider 2 variants in addition to the original version and illustrate them in Figure 4. The first variant, DyGFormer-separate, computes the temporal representations of the source and destination nodes separately. The goal of this variant is to rule out the effect of cross-sequence attention. The second variant, DyGDecoder, computes the source and destination node representations separately and in an autoregressive manner as described in Section 3.2.2. In DyGDecoder, we prepend a learnable "beginning of sequence" embedding to the input feature sequence matrices of the source and destination nodes, signaling the start of the sequences and enabling the transformer to count the sequence lengths (Kazemnejad et al., 2024). Similar to our approach for TGAT, we modify the time encoders of all three DyGFormer versions.

To verify that any differences between the sinusoidal and linear time encoders are not solely due to standardization, we also introduce a sinusoidal-scale time encoder. This encoder employs the same scaling strategy as the linear time encoder to the time differences but encodes them with sinusoidal functions, allowing us to isolate the impact of scaling from the underlying encoding technique.

## 4 Experiments

### 4.1 Setup

#### 4.1.1 Data

For our main experiments, we use six standard dynamic graph learning benchmark datasets: UCI (Panzarasa et al., 2009), Wikipedia (Kumar et al., 2019), Enron (Shetty & Adibi, 2004), Reddit (Kumar et al., 2019), LastFM (Kumar et al., 2019) and US Legis (Fowler, 2006; Huang et al., 2020). We list the details of the datasets in Appendix C. Following the setup of the unified dynamic graph learning library, DyGLib (Yu

Table 1: Test AP (random NS) across six datasets. The best performance for each model variant is highlighted in **bold** and the second best performance is highlighted by underline. Counts of dataset wins are also **boldfaced**, and in every model variant the linear time encoder secures the most wins.

| Test AP (Random NS) | Time Encoder | UCI | Wikipedia | Enron | Reddit | LastFM | US Legis | # Wins |
|---|---|---|---|---|---|---|---|---|
| TGAT | Sinusoidal | 80.27±0.42 | 97.00±0.15 | 72.07±1.10 | **98.55**±0.01 | 75.82±0.27 | 67.42±1.40 | 1 |
| | Sinusoidal-scale | 77.59±3.57 | 95.58±0.98 | 78.45±1.88 | 98.43±0.02 | 82.70±0.15 | **69.01**±1.86 | 1 |
| | Linear | **95.41**±0.06 | **98.21**±0.02 | **82.31**±0.08 | 98.52±0.02 | **83.10**±0.12 | 68.22±1.79 | **4** |
| DyGFormer | Sinusoidal | **96.01**±0.10 | 99.04±0.02 | **93.63**±0.11 | 99.20±0.01 | 93.68±0.04 | 70.12±0.60 | 2 |
| | Sinusoidal-scale | 84.76±6.69 | 99.15±0.08 | 93.19±0.21 | 99.25±0.01 | 94.33±0.05 | 70.08±0.64 | 0 |
| | Linear | 96.00±0.42 | **99.19**±0.02 | 93.29±0.15 | **99.28**±0.01 | **94.34**±0.04 | **70.44**±0.32 | **4** |
| DyGFormer-separate | Sinusoidal | **96.15**±0.02 | 99.04±0.02 | **93.49**±0.15 | 99.24±0.01 | 93.69±0.07 | 68.03±1.11 | 2 |
| | Sinusoidal-scale | 87.77±15.75 | 99.15±0.07 | 93.33±0.14 | 99.29±0.01 | **94.30**±0.11 | 69.36±1.84 | 1 |
| | Linear | 95.88±0.76 | **99.24**±0.02 | **93.49**±0.26 | **99.30**±0.01 | **94.30**±0.14 | **70.98**±1.22 | **5** |
| DyGDecoder | Sinusoidal | 96.11±0.06 | 99.03±0.02 | 93.53±0.04 | 99.25±0.01 | 94.21±0.03 | 69.39±1.06 | 0 |
| | Sinusoidal-scale | 94.25±1.29 | 98.93±0.14 | 93.62±0.16 | 99.30±0.04 | 94.32±0.13 | 69.52±0.86 | 0 |
| | Linear | **97.38**±0.06 | **99.21**±0.01 | **93.86**±0.18 | **99.32**±0.01 | **94.44**±0.03 | **70.12**±1.11 | **6** |

et al., 2023), we split the time span of an entire dataset into 70%/15%/15% for train/validation/test. We denote the starting time points for the validation and test splits by $t_{\text{val}}$ and $t_{\text{test}}$ where $0 < t_{\text{val}} < t_{\text{test}} < T$.

### 4.1.2 Negative Edge Sampling

As in DyGLib, we sample an equal amount of negative edges as the positive edges for the training and evaluation of future link prediction. We adopt two negative sampling (NS) strategies for evaluation: random NS and historical NS (Poursafaei et al., 2022). Let $\mathcal{E}(\check{t}, \hat{t}) := \{(i, j, t) | (i, j, t) \in \mathcal{E}, \check{t} \leq t < \hat{t}\}$ denote the set of edges that occur within the time range of $[\check{t}, \hat{t})$ and $\mathcal{P}(\check{t}, \hat{t}) := \{(i, j) | (i, j, t) \in \mathcal{E}(\check{t}, \hat{t})\}$ denote the node pairs that interact within the range. Given a batch of positive test edges $\mathcal{E}(\check{t}, \hat{t})$ where $t_{\text{test}} \leq \check{t} < \hat{t} < T$, random NS substitutes the node pairs of the positive edges by sampling from all possible node pairs uniformly: $\mathcal{E}_{\text{rand}}^-(\check{t}, \hat{t}) = \{(u, v, t) | u, v \sim \text{Unif}(\mathcal{V}), (i, j, t) \in \mathcal{E}(\check{t}, \hat{t})\}$. Historical NS draws node pairs that do not interact in the given batch of positive edges but have interacted in the past for substitution: $\mathcal{E}_{\text{hist}}^-(\check{t}, \hat{t}) = \{(u, v, t) | (u, v) \in \mathcal{P}(0, \check{t}), (u, v) \notin \mathcal{P}(\check{t}, \hat{t}), (i, j, t) \in \mathcal{E}(\check{t}, \hat{t})\}$. We train under random NS and do model selection under random and historical NS for evaluation under random and historical NS, respectively.

### 4.1.3 Evaluation Metric and Hyper-Parameters

We follow DyGLib to use the average precision (AP) as the evaluation metric, set the batch size to 200, and use the Adam optimizer (Kingma, 2014) with a learning rate of 0.0001. The sinusoidal time encoding dimension $d_T$ is set to 100. We also set the linear time encoding dimension to 100 for TGAT but set it to 1 for DyGFormer variants since the time encoder is followed by a linear projection to a temporal feature channel of dimension $d_{\text{ch}}$, effectively resulting in a $d_{\text{ch}}$-dimensional linear time encoder no matter what $d_T$ is.

We use the same other hyper-parameters for training the original DyGFormer under random NS as the ones in DyGLib since they are well-tuned. For the LastFM dataset, we directly apply the hyper-parameters specified in DyGLib as well due to the scale of the dataset. As for other cases, we search the hyper-parameters of each method on each dataset. The search space for TGAT is the dropout rate among $\{0.1, 0.3, 0.5\}$. The search space for DyGFormer variants is the combination of channel dimension $d_{\text{ch}} \in \{30, 50\}$ and the dropout rate among $\{0.1, 0.3, 0.5\}$. The best hyper-parameters are determined based on the validation AP score. Following Poursafaei et al. (2022) and DyGLib, we report the average results and standard deviation over five runs.

Table 2: Test AP (historical NS) across six datasets. The best performance for each model variant is highlighted in **bold** and the second best performance is highlighted by underline. Counts of dataset wins are also **boldfaced**, and in every model variant the linear time encoder secures the most wins.

| Test AP (Historical NS) | Time Encoder | UCI | Wikipedia | Enron | Reddit | LastFM | US Legis | # Wins |
|---|---|---|---|---|---|---|---|---|
| TGAT | Sinusoidal | 68.94$_{\pm0.58}$ | 88.02$_{\pm0.26}$ | 64.82$_{\pm2.31}$ | 79.62$_{\pm0.22}$ | 76.52$_{\pm0.77}$ | **84.59**$_{\pm4.39}$ | 1 |
| | Sinusoidal-scale | 41.84$_{\pm5.84}$ | 87.49$_{\pm0.64}$ | 78.40$_{\pm0.74}$ | 79.38$_{\pm0.36}$ | 80.99$_{\pm2.05}$ | 81.96$_{\pm2.51}$ | 0 |
| | Linear | **91.42**$_{\pm0.27}$ | **92.02**$_{\pm0.14}$ | **80.14**$_{\pm0.29}$ | **79.70**$_{\pm0.20}$ | **81.52**$_{\pm0.58}$ | 81.64$_{\pm4.21}$ | **5** |
| DyGFormer | Sinusoidal | 82.13$_{\pm0.75}$ | 83.81$_{\pm0.81}$ | 78.13$_{\pm0.24}$ | **83.11**$_{\pm0.64}$ | 82.83$_{\pm0.20}$ | 84.58$_{\pm1.24}$ | 1 |
| | Sinusoidal-scale | 80.29$_{\pm6.54}$ | 82.49$_{\pm3.11}$ | 75.32$_{\pm2.19}$ | 82.46$_{\pm0.55}$ | 83.79$_{\pm0.53}$ | **87.07**$_{\pm1.40}$ | 1 |
| | Linear | **89.41**$_{\pm2.63}$ | **86.23**$_{\pm0.85}$ | **78.24**$_{\pm1.24}$ | 82.31$_{\pm0.90}$ | **84.43**$_{\pm0.14}$ | 85.89$_{\pm1.36}$ | **4** |
| DyGFormer-separate | Sinusoidal | 82.42$_{\pm0.51}$ | 86.25$_{\pm0.89}$ | 75.70$_{\pm1.95}$ | 82.60$_{\pm0.73}$ | 82.70$_{\pm0.40}$ | 85.33$_{\pm2.58}$ | 0 |
| | Sinusoidal-scale | 80.92$_{\pm6.76}$ | 84.03$_{\pm0.80}$ | 76.87$_{\pm1.10}$ | 83.05$_{\pm1.42}$ | 83.90$_{\pm0.46}$ | 82.54$_{\pm3.29}$ | 0 |
| | Linear | **83.09**$_{\pm4.03}$ | **88.89**$_{\pm0.69}$ | **80.52**$_{\pm0.63}$ | **83.82**$_{\pm1.18}$ | **84.26**$_{\pm0.55}$ | **89.44**$_{\pm1.73}$ | **6** |
| DyGDecoder | Sinusoidal | 84.49$_{\pm0.61}$ | 87.01$_{\pm0.42}$ | 79.75$_{\pm0.19}$ | 84.65$_{\pm0.47}$ | **84.70**$_{\pm0.18}$ | **84.74**$_{\pm4.77}$ | 2 |
| | Sinusoidal-scale | 82.27$_{\pm1.91}$ | 86.27$_{\pm0.28}$ | 81.38$_{\pm0.30}$ | **85.09**$_{\pm0.52}$ | 84.57$_{\pm0.39}$ | 81.75$_{\pm4.17}$ | 1 |
| | Linear | **87.08**$_{\pm0.89}$ | **88.29**$_{\pm0.44}$ | **82.00**$_{\pm0.50}$ | 85.03$_{\pm0.55}$ | 84.41$_{\pm0.49}$ | 83.79$_{\pm2.12}$ | **3** |

## 4.2 Results and Discussions

### 4.2.1 Cosine vs. Sine-Cosine Pairs

TGAT and DyGFormer implement sinusoidal time encoders using only cosines, while their respective papers describe them as sine-cosine pairs. To investigate this, we first compare the performance of sinusoidal time encoders with all cosines and with sine-cosine pairs. The results are presented in Appendix D Table 4. We find that there is no significant difference in performance between the two versions. Therefore, we simply adopt the version with all cosines as implemented by TGAT and DyGFormer for the remainder of our study.

### 4.2.2 Evaluation Under Random NS

We compare sinusoidal, sinusoidal-scale, and linear time encoders under random NS and present the results in Table 1. The linear time encoder outperforms the others in 19 out of 24 model-dataset combinations, while the sinusoidal and sinusoidal-scale encoders perform best in 5 and 2 combinations, respectively. The largest improvements in test AP scores when switching from sinusoidal to linear are 15.14 and 10.24 for TGAT on UCI and Enron, respectively. The most notable performance drop is a modest decrease of 0.34 in the AP score for DyGFormer on Enron. When comparing linear and sinusoidal-scale time encoders, which both encode the same scaled time difference values using different functions, the linear encoder outperforms the sinusoidal-scale encoder in 22 out of 24 cases. This suggests that allowing self-attention to learn patterns relevant to time spans directly from the linear encodings is more effective than relying on pre-encoded sinusoidal functions.

Regarding the comparison between sinusoidal and sinusoidal-scale encoders, we observe that scaling consistently degrades performances on UCI, which is not the case for the other datasets. To investigate further, we analyze the inter-event times in UCI. For the source and destination nodes of each edge, we calculate the waiting times since their last interaction (see Appendix D.2 Table 5). We find that the waiting times increase significantly during the test split, which may hinder the sinusoidal-scale encoder's performance, as it relies on time difference statistics in the train split for scaling. Despite the negative impact of scaling under distribution shift, the linear time encoder is able to recover its performance. This echoes findings in the literature on relative positional encoding in language models (Press et al., 2022) which discovered that the linear function of positional differences shows strong length generalization.

### 4.2.3 Evaluation Under Historical NS

For evaluation under historical NS, we use the validation AP scores under historical NS for model selection. We present the results in Table 2. Among all 24 model-dataset combinations, the linear time encoder is

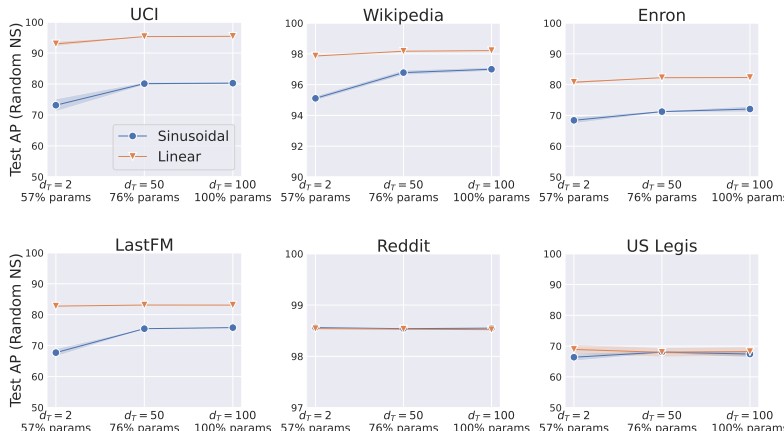

Figure 5: Test AP (random NS) vs. $d_T$ and the percentage of the number of parameters relative to $d_T = 100$ in TGAT.

the best in 18 instances while the sinusoidal and sinusoidal-scale encoders are the best in 4 and 2 instances, respectively.

The linear time encoder consistently outperforms the sinusoidal and sinusoidal-scale time encoder on UCI, Wikipedia, and Enron. The most significant differences caused by switching the time encoder from sinusoidal to linear are the increases in average test AP scores on UCI and Wikipedia from 68.94 to 91.42 and from 64.82 to 80.14 with TGAT. Other notable performance enhancements with DyGFormer variants include the 7.28 and 4.82 increases with DyGFormer on UCI and DyGFormer-separate on Enron.

We observe that the linear time encoder does not achieve the best performance on US Legis under historical NS evaluation, winning only with DyGFormer-separate. This is likely because sinusoidal time encoders are sufficient for this dataset's coarse-grained temporal granularity. US Legis is a discrete-time dynamic graph with only 12 evenly spaced time steps in the units of congressional sessions, resulting in just 12 possible time differences: $\{0, 1, ..., 11\}$. In such a setting, a simple one-dimensional sinusoidal encoding like $\cos(\frac{\pi}{11}\Delta t)$ can map all time differences to unique values in $[-1, 1]$ without collisions. In contrast, applying this approach to continuous-time datasets where time differences span up to $10^6$ seconds would lead to collisions due to limited numerical precision (e.g. $\cos(\frac{\pi}{10^6} \cdot 0) \approx \cos(\frac{\pi}{10^6} \cdot 85) \approx 1$ in float32). This example illustrates why encoding time differences is simpler in US Legis and why sinusoidal encodings are sufficient, while the advantage of the linear time encoder's injective property emerges in continuous-time scenarios.

## 4.3 Additional Analyses

### 4.3.1 Reducing Temporal Feature Dimension

In the synthetic task experiments in Section 3.2.3, we observe that transformers with a low-dimensional linear time encoder can effectively learn the synthetic task. We explore whether the same holds for realistic dynamic graph datasets.

In TGAT, the original time encoding dimension $d_T$ is 100. We reduce $d_T$ to be 50 and 2 for both sinusoidal and linear time encodings and plot the results in Figure 5. The most significant performance decline with the sinusoidal time encoder occurs on LastFM, where the AP score decreases by 8.08. In comparison, the largest performance drop with the linear time encoder is only 2.44 on UCI. On all datasets except Reddit, linear time encoder with $d_T = 2$ outperforms sinusoidal time encoder with $d_T = 100$, while saving 43% of parameters.

We conduct similar experiments with DyGFormer. In the original DyGFormer with sinusoidal time encoders, the dimension for each channel $d_{\mathrm{ch}}$ is 50. For UCI, Enron, LastFM, and US Legis, we also set $d_{\mathrm{ch}} = 50$ for DyGFormer with linear time encoders as determined in the hyper-parameter selection phase. On these

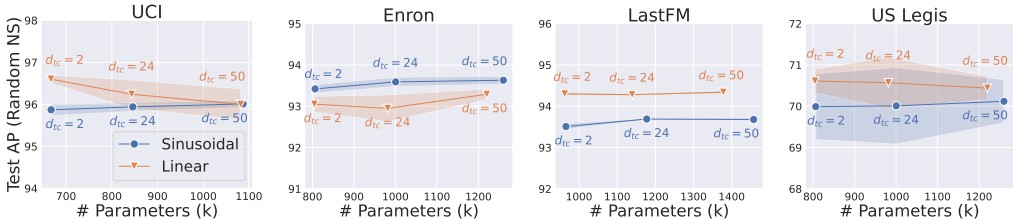

Figure 6: Test AP (random NS) under varying temporal feature channel dimension $d_{tc}$ and number of parameters in DyGFormer.

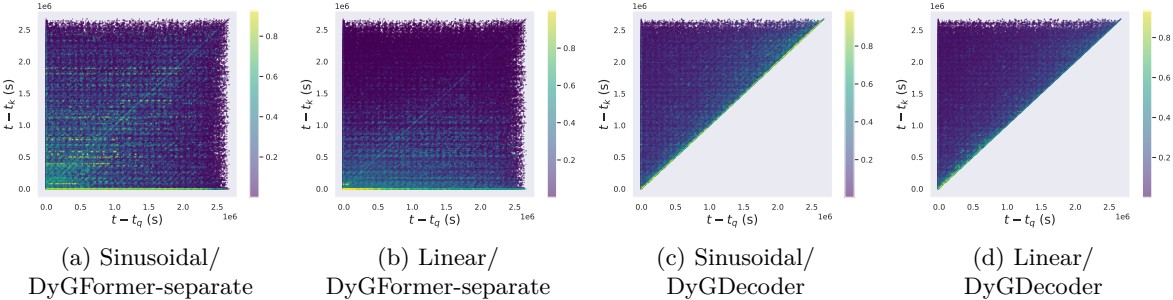

(a) Sinusoidal/ DyGFormer-separate    (b) Linear/ DyGFormer-separate    (c) Sinusoidal/ DyGDecoder    (d) Linear/ DyGDecoder

Figure 7: Attention scores (expressed in colors) of DyGFormer-separate and DyGDecoder with the sinusoidal and linear time encoders vs. $t - t_k$ vs. $t - t_q$ of the historical edge event sequences of Reddit's source nodes.

datasets, we experiment with reducing solely the dimension of the temporal feature channel to 24 and 2 while the dimensions for the other channels are fixed at 50. We separately denote the temporal feature channel dimension by $d_{tc}$ since it is different from the dimensions for the other channels $d_{ch}$ in this auxiliary experiment. The results, shown in Figure 6, reveal that while performances decrease on Enron and LastFM with both encoders as $d_{tc}$ is reduced, DyGFormer with a linear time encoder improves on UCI and US Legis, whereas the sinusoidal encoder continues to decline. Overall, these experiments confirm that linear time encoders are more parameter-efficient.

### 4.3.2 Attention Score Analysis

Attention scores are useful for understanding how edge event representations aggregate information. We choose to visualize the attention score patterns in the last layers of DyGFormer-separate and DyGDecoder since they rule out the effect of cross-sequence attention.

We sample $K$ target edges from the test set. For each edge $(i, j, t)$, we retrieve the historical edge event sequences of node $i$ and $j$. For each sequence, we record the attention score between a query vector and a key vector alongside their encoded time differences, $t - t_q$ and $t - t_k$. We then plot the attention scores and time differences for all $K$ historical edge event sequences of the source nodes and repeat the process for the destination nodes. Figure 7 shows the visualization results for the source nodes of Reddit. The other results are presented in Appendix D.7.

Figure 7a and 7b reveal distinct attention patterns of DyGFormer-separate on Reddit when different time encoders are used. With the linear encoder, the attention scores are negatively correlated with the elapsed time since the key edge events, so the colors smoothly transition along $t - t_k$. In contrast, the sinusoidal encoder produces scattered points of high attention scores separated by low-score regions, likely due to the time encoder's periodicity. However, such stark differences are absent across the other datasets. Regardless of the time encoder paired with DyGFormer-separate, attention scores tend to decline as $t - t_k$ increases, implying that the freshness of past interactions is typically the dominant factor.

Regarding the differences between the visualization results of DyGFormer-separate and DyGDecoder, the most notable one is that no attention scores are recorded in DyGDecoder for $t - t_q > t - t_k$. This arises

from its autoregressive attention mechanism. Another interesting distinction is that DyGFormer-separate's highest attention scores tend to be close to the line $t - t_k = 0$ while DyGDecoder's are close to the line $t - t_k = t - t_q$. This difference likely arises from their distinct pooling strategies. DyGDecoder uses the final event representation (i.e., the target edge event) as the sequence representation, which aggregates information from preceding events. Consequently, it learns to attend more strongly to key vectors where $t_k$ is close to $t_q$. In contrast, DyGFormer-separate applies mean pooling over all edge event representations to form a sequence representation, so even a query vector corresponding to an early event allocates most of its attention to the latest events (where $t_k$ is closer to $t$) because those events are more influential.

In summary, our experimental results demonstrate that linear time encoders mostly outperform sinusoidal encoders when paired with TGAT and DyGFormer. Moreover, self-attention can effectively learn temporal patterns from linear time encodings with far fewer dimensions. While both time encoders can be used together in practice, as in Time2vec (Kazemi et al., 2019) used in time series modeling (Shukla & Marlin, 2021), our findings highlight that the linear encoder offers advantages in reducing model complexity. For large-scale temporal network applications, adopting linear time encodings can significantly streamline the architecture by avoiding high dimensional sinusoidal encodings, as used in TGAT and DyGFormer.

## 5 Related Work

Dynamic graphs (Holme & Saramäki, 2012; Longa et al., 2023) are evolving graphs associated with temporal information. There are two classes of dynamic graph learning methods: discrete-time methods (Pareja et al., 2020; Sankar et al., 2020; You et al., 2022), which discretize dynamic graphs into static snapshots and lose the temporal order within each snapshot, and continuous-time methods (Kumar et al., 2019; Trivedi et al., 2019; Rossi et al., 2020; da Xu et al., 2020; Wang et al., 2021c; Cong et al., 2023; Yu et al., 2023), which directly model temporal dynamics and are more versatile. This work focuses on the latter.

Continuous-time methods can be divided into memory-based and non-memory-based methods. Memory-based methods (Trivedi et al., 2019; Kumar et al., 2019; Rossi et al., 2020) maintain a memory for each node, updating it whenever an event related to the node occurs. In contrast, non-memory-based methods retrieve a node's historical information as needed, allowing for more expressive models to compute its temporal representation. Non-memory-based methods can be further subdivided into graph-based and sequence-based approaches. TGAT (da Xu et al., 2020) and TCL (Wang et al., 2021b) belong to the graph-based category, as they aggregate higher-order temporal neighbor information. On the other hand, DyGFormer (Yu et al., 2023) and GraphMixer (Cong et al., 2023) are sequence-based approaches because they only consider first-order temporal neighbors. CAWN (Wang et al., 2021c) sits between the two approaches, as it utilizes a recurrent neural network (RNN) (Rumelhart et al., 1986) to learn representations from causal anonymous random walks over the graph. It is worth noting that TCL uses linear functions to represent time values, which is similar to the linear time encoder that we present in Section 3. However, TCL does not provide a justification for the design choice nor compare it with sinusoidal encodings. Aside from methods, evaluation is also a crucial component of dynamic graph learning. Poursafaei et al. (2022) introduce historical and inductive negative sampling to test if future link prediction models simply memorize past edges. TGB (Huang et al., 2024; Gastinger et al., 2024) provides a standardized package that includes datasets, data loaders, and evaluators. DyGLib (Yu et al., 2023) is a unified library for dynamic graph learning that contains implementations of various methods, data processing, and evaluation.

The sinusoidal time encoder (Xu et al., 2019; da Xu et al., 2020) resembles the positional encoding in Transformer (Waswani et al., 2017), which uses sinusoidal functions to encode discrete language token positions. Subsequently, there have been a series of development in relative positional encodings (RPE) in language modeling (Shaw et al., 2018; Raffel et al., 2020; Press et al., 2022; Chi et al., 2022; Su et al., 2024), which inject relative positional information to attention scores after computing the inner products between key and query vectors. We do not consider language model RPEs in this paper for two reasons. Firstly, language model RPEs have not been directly applied to dynamic graph learning to the best of our knowledge. Secondly, there is only a single position for each word token in language modeling while DyGFormer uses a patching technique to merge multiple edge events into a single "token". Therefore, each "token" in DyGFormer may be associated with multiple time positions and it is unclear how to compute the relative positional infor-

mation between two tokens. This would not be a problem for sinusoidal and linear time encodings because they can be pre-computed and then patched into the same vector before self-attention. Time2Vec (Kazemi et al., 2019) is a time encoding technique used in irregular time series modeling (Shukla & Marlin, 2021) that combines a linear time feature with multiple periodic time features. The key difference between Time2Vec's linear time feature and our linear time encoder is that our encoder standardizes time values, which we found significantly benefits optimization efficiency and improves results in the early stages of our experiments. This work focuses on the comparison between linear and sinusoidal time encoders to isolate the effects of their respective functional forms. We leave a thorough exploration of their combination to future work.

## 6    Conclusion

In this study, we present the linear time encoder as an effective alternative to the widely-used sinusoidal time encoder for attention-based dynamic graph learning models. Our theoretical and experimental findings demonstrate that the linear time encoder not only allows the self-attention mechanism to capture temporal relationships but also outperforms sinusoidal time encoders in most scenarios, particularly in terms of predictive accuracy and parameter savings. While both encoders can be used together simultaneously, our study sheds lights on the advantages of linear time features, which are overlooked in modern dynamic graph models. Looking ahead, exploring the possibility of adapting language model RPEs to continuous-time dynamic graphs is an exciting direction, as they have shown better generalization performances in the language domain. Ultimately, these advancements will enable more effective dynamic graph models, driving improvements in applications such as recommender systems and traffic forecasting.

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

# A   Details of Attention-Based Dynamic Graph Learning Models

## A.1   TGAT

TGAT (da Xu et al., 2020) is a graph-based dynamic graph learning model. We describe the operation of the $l$'th layer of TGAT when computing the representation of node $i$ at target time $t$, denoted by $\tilde{\mathbf{h}}_i^{(l)}(t) \in \mathbb{R}^d$. The initial temporal representation of node $i$ is its raw attributes, i.e. $\tilde{\mathbf{h}}_i^{(0)}(t) = \mathbf{x}_i$. To compute the next layer of node $i$'s temporal representation, TGAT uses self-attention where node $i$ acts as the query and its temporal neighbors act as the keys and values. For simplicity, we label the temporal neighbors of node $i$ as $\mathcal{N}_i^t = \{(v_1, t_1), (v_2, t_2), \dots (v_{|\mathcal{N}_i^t|}, t_{|\mathcal{N}_i^t|})\}$. Let $\Phi : \mathcal{I}_T \to \mathbb{R}^{d_T}$ denote the time encoder and $\mathbf{W}_Q, \mathbf{W}_K, \mathbf{W}_V \in \mathbb{R}^{(d+d_T) \times d_h}$ be the query, key, and value weight matrices. For each node $v$ and time $t'$ involved, the input to the self-attention consists of the $l-1$'th layer representation $\tilde{\mathbf{h}}_v^{(l-1)}(t')$ and the encoded time difference $\Phi(t - t')$. In specific, the query vector corresponding to node $i$ at time $t$, $\mathbf{q}_i(t)$, is computed as:

$$\mathbf{q}_i(t) = \mathbf{W}_Q^\top \begin{bmatrix} \tilde{\mathbf{h}}_i^{(l-1)}(t) \\ \Phi(0) \end{bmatrix} \in \mathbb{R}^{d_h}$$

The time difference input to $\Phi$ is 0 because the difference between the target time $t$ and itself is 0. The input to the key and value weight matrices are the temporal neighbor features $\mathbf{N}_i(t)$:

$$\mathbf{N}_i(t) = \begin{bmatrix} \tilde{\mathbf{h}}_{v_1}^{(l-1)}(t_1)^\top & \Phi(t - t_1)^\top \\ \tilde{\mathbf{h}}_{v_2}^{(l-1)}(t_2)^\top & \Phi(t - t_2)^\top \\ \vdots & \\ \tilde{\mathbf{h}}_{v_{|\mathcal{N}_i^t|}}^{(l-1)}(t_{|\mathcal{N}_i^t|})^\top & \Phi(t - t_{|\mathcal{N}_i^t|})^\top \end{bmatrix} \in \mathbb{R}^{|\mathcal{N}_i^t| \times (d + d_T)}$$

The remaining part of self-attention is carried out as follows to aggregate the neighbor information to a vector $\mathbf{h}_i(t)$:

$$\mathbf{K}_i(t) = \mathbf{N}_i(t)\mathbf{W}_K \in \mathbb{R}^{|\mathcal{N}_i^t| \times d_h}$$
$$\mathbf{V}_i(t) = \mathbf{N}_i(t)\mathbf{W}_V \in \mathbb{R}^{|\mathcal{N}_i^t| \times d_h}$$
$$\mathbf{h}_i(t) = \text{Attn}(\mathbf{q}_i(t), \mathbf{K}_i(t), \mathbf{V}_i(t))$$
$$= \mathbf{V}_i(t)^\top \text{Softmax}(\frac{\mathbf{K}_i(t)\mathbf{q}_i(t)}{\sqrt{d_h}}) \in \mathbb{R}^{d_h}$$

Finally, TGAT concatenates $\mathbf{h}_i(t)$ with node $i$'s raw attributes and forwards it through an MLP to obtain its $l$'th-layer temporal representation $\tilde{\mathbf{h}}_i^{(l)}(t)$.

$$\tilde{\mathbf{h}}_i^{(l)}(t) = \mathbf{W}_1 \text{ReLU}(\mathbf{W}_0 \begin{bmatrix} \mathbf{h}_i(t) \\ \mathbf{x}_i \end{bmatrix} + \mathbf{b}_0) + \mathbf{b}_1$$

where $\mathbf{W}_0 \in \mathbb{R}^{d_f \times (d_h + d_V)}, \mathbf{W}_1 \in \mathbb{R}^{d \times d_f}, \mathbf{b}_0 \in \mathbb{R}^{d_f}, \mathbf{b}_1 \in \mathbb{R}^d$ are the MLP parameters. TGAT can be extended to make use of edge features corresponding to each temporal neighbor by concatenating them to $\mathbf{N}_i(t)$.

## A.2   DyGFormer

DyGFormer (Yu et al., 2023) is a sequence-based dynamic graph learning model. To compute the temporal representation of node $i$, the target edge $(i, j, t)$ and the historical edge interactions involving $i$, $\mathcal{S}_i^t$, are

retrieved and sorted by the timestamps in ascending order. We denote the edge collection by $\tilde{\mathcal{S}}_i^t := \mathcal{S}_i^t \cup \{(i, j, t)\}$. Then, the features corresponding to those edges are collected and arranged in the same order. There are four channels of features: temporal features, neighbor node features, edge features, and neighbor co-occurrence features. The temporal feature matrix $\mathbf{X}_{i,T}^t \in \mathbb{R}^{|\tilde{\mathcal{S}}_i^t| \times d_T}$ consists of the time encoding $\Phi(t - t') \in \mathbb{R}^{d_T}$ of each edge that has a timestamp $t'$. Similarly, the neighbor node feature matrix $\mathbf{X}_{i,V}^t \in \mathbb{R}^{|\tilde{\mathcal{S}}_i^t| \times d_V}$ and edge feature matrix $\mathbf{X}_{i,E}^t \in \mathbb{R}^{|\tilde{\mathcal{S}}_i^t| \times d_E}$ contain the raw neighbor node attributes and edge attributes of each edge. Note that since the target edge has not occurred, its edge attributes are filled with a zero vector.

The use of neighbor co-occurrence features is a key technique in DyGFormer to strengthen the modeling of the correlation between two nodes, $i$ and $j$, of the target edge $(i, j, t)$ for better link prediction. Given $\tilde{\mathcal{S}}_i^t$ and $\tilde{\mathcal{S}}_j^t$, the occurrence frequencies of each neighbor in the histories of nodes $i$ and $j$ are encoded into the neighbor co-occurrence matrices $\mathbf{C}_i^t \in \mathbb{R}^{|\tilde{\mathcal{S}}_i^t| \times 2}$ and $\mathbf{C}_j^t \in \mathbb{R}^{|\tilde{\mathcal{S}}_j^t| \times 2}$. In specific, the $m$'th row of $\mathbf{C}_*^t$ contains the occurrence frequencies of the $m$'th temporal neighbor of node $*$ in node $i$ and $j$'s histories where $*$ can be either $i$ or $j$. For example, if the temporal neighbors of $i$ and $j$ (including the target edge) are $\{(u, t_1), (v, t_2), (w, t_3), (j, t)\}$ and $\{(v, t_4), (u, t_5), (v, t_6), (v, t_7), (i, t)\}$, we can count the frequencies of $u, v, w, i$ and $j$'s appearances in $i/j$'s histories as 1/1, 1/3, 1/0, 0/1, 1/0. Then, the neighbor co-occurrence matrices $\mathbf{C}_i^t$ and $\mathbf{C}_j^t$ can be written as:

$$\mathbf{C}_i^t = \begin{bmatrix} 1 & 1 \\ 1 & 3 \\ 1 & 0 \\ 1 & 0 \end{bmatrix}, \mathbf{C}_j^t = \begin{bmatrix} 1 & 3 \\ 1 & 1 \\ 1 & 3 \\ 1 & 3 \\ 0 & 1 \end{bmatrix}$$

Later, the neighbor co-occurrence matrices are transformed by an MLP $f_C$ into neighbor co-occurrence features of dimension $d_C$:

$$\mathbf{X}_{*,C}^t = f_C((\mathbf{C}_*^t)_{:,1}) + f_C((\mathbf{C}_*^t)_{:,2}) \in \mathbb{R}^{|\tilde{\mathcal{S}}_*^t| \times d_C}$$

where $*$ can be either $i$ or $j$.

After collecting the four channels of feature matrices, DyGFormer applies a patching technique, which essentially reshapes the matrices, to allow the transformer to capture longer-range information without increasing the computational cost. Let $P$ denote the patch size. Given a sequence of channel $*$ features $\mathbf{X}_{i,*}^t \in \mathbb{R}^{|\tilde{\mathcal{S}}_i^t| \times d_*}$, the sequence is divided into $l_i^t := \lceil |\tilde{\mathcal{S}}_i^t| / P \rceil$ patches and each patch is reshaped into a single $Pd_*$-dimensional vector, resulting in a shorter sequence of patches $\mathbf{P}_{i,*}^t \in \mathbb{R}^{l_i^t \times P d_*}$. If we fix the resulting sequence length $l_i^t$ and increase the patch size $P$, the transformer can see through a longer history. The tradeoff is that the information of historical edge interactions within the same patch will be aggregated via a simple concatenation instead of the attention mechanism. The patching technique is applied to all channels of features and produces the patch sequences for node $i$: $\mathbf{P}_{i,V}^t \in \mathbb{R}^{l_i^t \times P d_V}, \mathbf{P}_{i,E}^t \in \mathbb{R}^{l_i^t \times P d_E}, \mathbf{P}_{i,T}^t \in \mathbb{R}^{l_i^t \times P d_T}, \mathbf{P}_{i,C}^t \in \mathbb{R}^{l_i^t \times P d_C}$. Afterward, each channel of the patch sequences is linearly transformed into the same dimension of $d_{\text{ch}}$ and concatenated, resulting in the final input to the transformer:

$$\mathbf{X}_i^t = f_1(\mathbf{P}_{i,V}^t) || f_2(\mathbf{P}_{i,E}^t) || f_3(\mathbf{P}_{i,T}^t) || f_4(\mathbf{P}_{i,C}^t) \in \mathbb{R}^{l_i^t \times 4 d_{\text{ch}}}$$

where $f_1, f_2, f_3$, and $f_4$ are linear transformations.

The feature matrices $\mathbf{X}_i^t$ and $\mathbf{X}_j^t$ corresponding to the two nodes of the target edge $(i, j, t)$ are concatenated in the sequence dimension:

$$\mathbf{X}^t = \begin{bmatrix} \mathbf{X}_i^t \\ \mathbf{X}_j^t \end{bmatrix} \in \mathbb{R}^{(l_i^t + l_j^t) \times 4 d_{\text{ch}}}$$

and forwarded through $L$ transformer layers to output $\mathbf{Z}^{(L)}(t) \in \mathbb{R}^{(l_i^t + l_j^t) \times d_h}$ where $d_h$ is the attention dimension.

DyGFormer sets the attention dimension $d_h$ of all layers to be the same as the input dimension, i.e. $d_h = 4 d_{\text{ch}}$. Let $\mathbf{Z}^{(l)}(t) \in \mathbb{R}^{(l_i^t + l_j^t) \times d_h}$ denote the $l$'th-layer sequence representation that is initialized as the input, i.e. $\mathbf{Z}^{(0)}(t) = \mathbf{X}^t$. $\mathbf{W}_Q^{(l)}, \mathbf{W}_K^{(l)}, \mathbf{W}_V^{(l)} \in \mathbb{R}^{d_h \times d_h}$ denote the $l$'th layer attention weight matrices, and $\mathbf{W}_1^{(l)} \in$

$\mathbb{R}^{d_h \times 4d_h}, \mathbf{W}_2^{(l)} \in \mathbb{R}^{4d_h \times d_h}, \mathbf{b}_1^{(l)} \in \mathbb{R}^{4d_h}, \mathbf{b}_2^{(l)} \in \mathbb{R}^{d_h}$ denote the $l'$th-layer MLP parameters. The $l'$th-layer transformer computation goes as:

$$\mathbf{Q}^{(l)} = \text{LayerNorm}(\mathbf{Z}^{(l-1)}(t))\mathbf{W}_Q^{(l)}$$

$$\mathbf{K}^{(l)} = \text{LayerNorm}(\mathbf{Z}^{(l-1)}(t))\mathbf{W}_K^{(l)}$$

$$\mathbf{V}^{(l)} = \text{LayerNorm}(\mathbf{Z}^{(l-1)}(t))\mathbf{W}_V^{(l)}$$

$$\mathbf{O}^{(l)} = \text{Attn}(\mathbf{Q}^{(l)}, \mathbf{K}^{(l)}, \mathbf{V}^{(l)}) = \text{Softmax}(\mathbf{Q}^{(l)}\mathbf{K}^{(l)\top}/\sqrt{d_h})\mathbf{V}^{(l)}$$

$$\mathbf{H}^{(l)} = \mathbf{O}^{(l)} + \mathbf{Z}^{(l-1)}(t)$$

$$\mathbf{B}_1^{(l)} = \begin{bmatrix} \mathbf{b}_1^{(l)\top} \\ \vdots \\ \mathbf{b}_1^{(l)\top} \end{bmatrix}_{(l_i^t + l_j^t) \times 4d_h} \qquad \mathbf{B}_2^{(l)} = \begin{bmatrix} \mathbf{b}_2^{(l)\top} \\ \vdots \\ \mathbf{b}_2^{(l)\top} \end{bmatrix}_{(l_i^t + l_j^t) \times d_h}$$

$$\mathbf{Y}^{(l)} = \text{GELU}(\text{LayerNorm}(\mathbf{H}^{(l)})\mathbf{W}_1^{(l)} + \mathbf{B}_1^{(l)})\mathbf{W}_2^{(l)} + \mathbf{B}_2^{(l)}$$

$$\mathbf{Z}^{(l)}(t) = \mathbf{H}^{(l)} + \mathbf{Y}^{(l)}$$

After $L$ layers of transformer computations, DyGFormer applies mean pooling followed by a linear transformation to the positions corresponding to $i$ and $j$ in $\mathbf{Z}^{(L)}(t)$ to get their respective temporal representations.

## B Proof of Proposition 3.1

**Proposition 3.1 (Restated).** *Let $\boldsymbol{x} = \{(\mathbf{x}_1, t_1), \ldots, (\mathbf{x}_M, t_M)\}$ be a set of events where each event $m$ contains a feature vector $\mathbf{x}_m \in \mathbb{R}^d$ and a timestamp $t_m \in \mathcal{I}_T$. Let $t$ be the target time. There exists a linear time encoder $\Phi : \mathcal{I}_T \to \mathbb{R}^{d_T}$, a query weight matrix $\mathbf{W}_Q \in \mathbb{R}^{(d_T+d) \times d_h}$, and a key weight matrix $\mathbf{W}_K \in \mathbb{R}^{(d_T+d) \times d_h}$ that can factor the time span $t_m - t_n$ between any two events $m, n$ into the attention score:*

$$\mathbf{q}_m = \mathbf{W}_Q^\top \begin{bmatrix} \Phi(t - t_m) \\ \mathbf{x}_m \end{bmatrix}, \, \mathbf{k}_n = \mathbf{W}_K^\top \begin{bmatrix} \Phi(t - t_n) \\ \mathbf{x}_n \end{bmatrix}$$

$$\langle \mathbf{q}_m, \mathbf{k}_n \rangle = h(t_m - t_n) + g(\mathbf{q}_m, \mathbf{k}_n)$$

*where $h : \mathbb{R} \to \mathbb{R}$ models the patterns relevant to the time span and $g : \mathbb{R}^{d_T+d} \times \mathbb{R}^{d_T+d} \to \mathbb{R}$ models other patterns between the events.*

*Proof.* We provide a construction of $\Phi$, $\mathbf{W}_Q$, and $\mathbf{W}_K$ that only requires specifying two dimensions of the parameters. We first construct the linear time encoder $\Phi$ as follows:

$$\Phi(t - t') = \begin{bmatrix} w_1(t - t') + b_1 \\ 1 \\ w_3(t - t') + b_3 \\ \vdots \\ w_{d_T}(t - t') + b_{d_T} \end{bmatrix}$$

where $w_1 \neq 0$, $w_2 = 0$, and $b_2 = 1$. There are no restrictions on other time encoder parameters: $b_1, b_3, b_4, \ldots b_{d_T}$ and $w_3, w_4, \ldots w_{d_T}$. Then, we construct the query matrix $\mathbf{W}_Q$ and the key matrix $\mathbf{W}_K$

as follow:

$$\mathbf{W}_Q = \begin{bmatrix} 1 & 0 & q_{1,3} & \cdots & q_{1,d_h} \\ 0 & -1 & q_{2,3} & \cdots & q_{2,d_h} \\ 0 & 0 & q_{3,3} & \cdots & q_{3,d_h} \\ \vdots & \vdots & \vdots & \ddots & \vdots \\ 0 & 0 & q_{d_T+d,3} & \cdots & q_{d_T+d,d_h} \end{bmatrix}_{(d_T+d)\times d_h} \qquad \mathbf{W}_K = \begin{bmatrix} 0 & 1 & k_{1,3} & \cdots & k_{1,d_h} \\ 1 & 0 & k_{2,3} & \cdots & k_{2,d_h} \\ 0 & 0 & k_{3,3} & \cdots & k_{3,d_h} \\ \vdots & \vdots & \vdots & \ddots & \vdots \\ 0 & 0 & k_{d_T+d,3} & \cdots & k_{d_T+d,d_h} \end{bmatrix}_{(d_T+d)\times d_h}$$

where only the first two columns are specified. Following the constructions, the query vector $\mathbf{q}_m$ representing event $m$ and the key vector $\mathbf{k}_n$ representing event $n$ have the form of:

$$\mathbf{q}_m = \mathbf{W}_Q^\top \begin{bmatrix} \Phi(t-t_m) \\ \mathbf{x}_m \end{bmatrix} = \begin{bmatrix} w_1(t-t_m)+b_1 \\ -1 \\ q_3 \\ \vdots \\ q_{d_h} \end{bmatrix}, \mathbf{k}_n = \mathbf{W}_K^\top \begin{bmatrix} \Phi(t-t_n) \\ \mathbf{x}_n \end{bmatrix} = \begin{bmatrix} 1 \\ w_1(t-t_n)+b_1 \\ k_3 \\ \vdots \\ k_{d_h} \end{bmatrix}$$

Therefore, the inner product between the two vectors will compute the time span between two events and factor it into the attention score:

$$\langle \mathbf{q}_m, \mathbf{k}_n \rangle = -w_1(t_m - t_n) + \sum_{i=3}^{d_h} q_i k_i = h(t_m - t_n) + g(\mathbf{q}_m, \mathbf{k}_n)$$

Note that this is just one way of constructing the linear time encoder and attention weights. There may exist numerous other possible implementations that achieve the same outcome. This proof is inspired by Kazemnejad et al. (2024) where they show that a two-layer autoregressive transformer without using positional encodings can recover the absolute and relative positions of tokens. The key difference between our setting and theirs is that our notion of distance between events is the time interval between two events instead of the number of "tokens" between them. $\qquad\square$

## C Dataset Details

We describe the datasets that we used for the experiments below and present the dataset statistics in Table 3:

- UCI (Panzarasa et al., 2009): An unattributed online communication network among University of California Irvine students.

- Wikipedia (Kumar et al., 2019): This dataset contains edits made to Wikipedia pages over one month where editors and pages are represented as nodes and timestamped posting requests are edges. Edge features are LIWC feature vectors (Pennebaker, 2001) of length 172 extracted from the edit texts.

- Enron (Shetty & Adibi, 2004): An unattributed temporal network of approximately 50,000 emails exchanged among employees of the Enron energy company over three years.

- Reddit (Kumar et al., 2019): This dataset consists of subreddit posts over one month with nodes representing users or posts and edges representing timestamped posting requests. Edge features are LIWC feature vectors (Pennebaker, 2001) of length 172 extracted from the edit texts.

- LastFM (Kumar et al., 2019): An unattributed interaction network where nodes represent users and songs, and edges denote instances of a user listening to a song. The dataset captures the listening behaviors of 1,000 users with the 1,000 most popular songs over the course of one month.

- US Legis (Fowler, 2006; Huang et al., 2020): A senate co-sponsorship graph documenting social interactions between U.S. Senate legislators with edge weights indicating how many times two legislators have co-sponsored a bill.

Table 3: Dataset statistics.

| Dataset | # Nodes | Total # Edges | Unique # Edges | Unique # Steps | Time Granularity | Duration |
|---------|---------|---------------|----------------|----------------|------------------|----------|
| UCI | 1899 | 59835 | 20296 | 58911 | Unix timestamp | 196 days |
| Wikipedia | 9227 | 157474 | 18257 | 152757 | Unix timestamp | 1 month |
| Enron | 184 | 125235 | 3125 | 22632 | Unix timestamp | 3 years |
| Reddit | 10984 | 672447 | 78516 | 669065 | Unix timestamp | 1 month |
| LastFM | 1980 | 1293103 | 154993 | 1283614 | Unix timestamp | 1 month |
| US Legis | 225 | 60396 | 26423 | 12 | Congresses | 12 congresses |

Table 4: A comparison between the sinusoidal time encoder using all cosine functions and one using sine-cosine pairs over six datasets. The better performance for each model variant is highlighted in **bold**. Same performances are highlighted by underline.

| Test AP (Random NS) | Time Encoder | UCI | Wikipedia | Enron | Reddit | LastFM | US Legis | # Wins |
|---------------------|--------------|-----|-----------|-------|--------|--------|----------|--------|
| TGAT | Cosine | $80.27_{\pm0.42}$ | $97.00_{\pm0.15}$ | $\mathbf{72.07}_{\pm1.10}$ | $\mathbf{98.55}_{\pm0.01}$ | $75.82_{\pm0.27}$ | $\mathbf{67.42}_{\pm1.40}$ | 3 |
|  | Sine-cosine | $\mathbf{81.48}_{\pm0.47}$ | $\mathbf{97.11}_{\pm0.14}$ | $70.94_{\pm2.12}$ | $98.54_{\pm0.02}$ | $\mathbf{76.05}_{\pm0.32}$ | $67.14_{\pm1.90}$ | 3 |
| DyGFormer | Cosine | $96.01_{\pm0.10}$ | $\underline{99.04}_{\pm0.02}$ | $\mathbf{93.63}_{\pm0.11}$ | $99.20_{\pm0.01}$ | $93.68_{\pm0.04}$ | $70.12_{\pm0.60}$ | 1 |
|  | Sine-cosine | $\mathbf{96.03}_{\pm0.06}$ | $\underline{99.04}_{\pm0.04}$ | $93.53_{\pm0.10}$ | $\mathbf{99.21}_{\pm0.01}$ | $\mathbf{93.73}_{\pm0.05}$ | $\mathbf{70.23}_{\pm0.72}$ | 4 |
| DyGFormer-separate | Cosine | $\mathbf{96.15}_{\pm0.02}$ | $\underline{99.04}_{\pm0.02}$ | $\mathbf{93.49}_{\pm0.15}$ | $\underline{99.24}_{\pm0.01}$ | $93.69_{\pm0.07}$ | $68.03_{\pm1.11}$ | 2 |
|  | Sine-cosine | $95.88_{\pm0.76}$ | $\underline{99.04}_{\pm0.02}$ | $93.42_{\pm0.32}$ | $\underline{99.24}_{\pm0.01}$ | $\mathbf{93.73}_{\pm0.07}$ | $\mathbf{69.25}_{\pm1.15}$ | 2 |
| DyGDecoder | Cosine | $\mathbf{96.11}_{\pm0.06}$ | $99.03_{\pm0.02}$ | $\mathbf{93.53}_{\pm0.04}$ | $99.25_{\pm0.01}$ | $94.21_{\pm0.03}$ | $\mathbf{69.39}_{\pm1.06}$ | 3 |
|  | Sine-cosine | $96.09_{\pm0.03}$ | $\mathbf{99.07}_{\pm0.02}$ | $93.38_{\pm0.32}$ | $\mathbf{99.26}_{\pm0.01}$ | $\mathbf{94.29}_{\pm0.04}$ | $68.46_{\pm2.21}$ | 3 |

# D Supplementary Materials For Experiments

## D.1 Cosine vs. Sine-Cosine Pairs

We compare sinusoidal encoders with all cosines as implemented in TGAT and DyGFormer and sinusoidal encoders with sine-cosine pairs as described in their corresponding papers to determine which to use in our study on dynamic graph datasets. We train TGAT, DyGFormer, DyGFormer-separate, and DyGDecoder using both versions of the time encoder and present the results in Table 4. The version with all cosines outperforms the sine-cosine pair encoder in 9 out of 24 model-dataset combinations, while the sine-cosine pair version is better in 12 combinations. The remaining 3 combinations result in tied average AP scores. Our findings indicate that there is no significant difference in performance between the two time encoders. Therefore, we simply adopt the sinusoidal time encoder with all cosines as implemented in TGAT and DyGFormer for the remaining of our empirical study.

## D.2 Waiting Time Statistics in UCI

In the main experimental results presented in Table 1, we observe that the sinusoidal-scale encoder consistently underperforms compared to the sinusoidal encoder on the UCI dataset. To investigate this, we analyze the dataset statistics, focusing on the waiting time between a node's current interaction and its previous one, with the results shown in Table 5. Our analysis reveals that the waiting time significantly increases during the test split. This increase may explain the underperformance of the sinusoidal-scale encoder, as its scaler relies on the empirical mean and standard deviation from the train split.

## D.3 Sinusoidal Time Encoding vs. Sinusoidal Positional Encoding

To highlight the importance of temporal information beyond mere sequential order, we conduct a separate experiment comparing sinusoidal time encodings with sinusoidal positional encodings. The latter encodes the positional indices of events rather than their time values. Table 6 presents the results. We observe that time encoding outperforms positional encoding in 10 out of 12 cases, indicating that temporal features are

Table 5: Statistics of the waiting time since the last edge interaction of source and destination nodes of each edge in UCI. We ignore the first appearance of a node since there are no previous interactions. The statistics are aggregated by train/validation/test splits.

| Waiting Time (Minutes) | Source Node | | Destination Node | |
|---|---|---|---|---|
| | Average | Median | Average | Median |
| Train | 266.90 | 3.97 | 415.55 | 9.95 |
| Validation | 414.49 | 4.60 | 717.26 | 11.16 |
| Test | 2129.37 | 26.25 | 4476.33 | 145.57 |

Table 6: A comparison between sinusoidal time encoding and sinusoidal positional encoding. The better performance for each model variant is highlighted in **bold**. The number in the parentheses is the performance change after switching from encoding time values to position indices.

| Test AP (Random NS) | Encoded Feature | UCI | Wikipedia | Enron | Reddit | LastFM | US Legis |
|---|---|---|---|---|---|---|---|
| TGAT | Time | $\mathbf{80.27}_{\pm 0.42}$ | $\mathbf{97.00}_{\pm 0.15}$ | $\mathbf{72.07}_{\pm 1.10}$ | $98.55_{\pm 0.01}$ | $\mathbf{75.82}_{\pm 0.27}$ | $\mathbf{67.42}_{\pm 1.40}$ |
| | Position | $79.09_{\pm 0.26}$ (-1.18) | $95.98_{\pm 0.10}$ (-1.02) | $65.61_{\pm 2.78}$ (-6.46) | $\mathbf{98.63}_{\pm 0.01}$ (+0.08) | $58.78_{\pm 0.21}$ (-17.04) | $50.11_{\pm 3.69}$ (-17.31) |
| DyGFormer | Time | $\mathbf{96.01}_{\pm 0.10}$ | $\mathbf{99.04}_{\pm 0.02}$ | $\mathbf{93.63}_{\pm 0.11}$ | $\mathbf{99.20}_{\pm 0.01}$ | $93.68_{\pm 0.04}$ | $\mathbf{70.12}_{\pm 0.60}$ |
| | Position | $95.94_{\pm 0.02}$ (-0.07) | $98.87_{\pm 0.03}$ (-0.17) | $92.97_{\pm 0.34}$ (-0.66) | $99.19_{\pm 0.01}$ (-0.01) | $\mathbf{94.26}_{\pm 0.01}$ (+0.58) | $66.58_{\pm 1.59}$ (-3.54) |

generally beneficial. Among all datasets, Reddit appears to be the least affected by the switch, with changes in AP scores below 0.1. This suggests that temporal features are less critical in Reddit, which aligns with our observations in Section 4.3.1 Figure 5.

### D.4 Training Time and Memory Usage

For the main experiments, we measure the training time per epoch and peak GPU memory usage across datasets and model variants and present them in Table 7 and 8. In general, models using the sinusoidal time encoder exhibit longer runtimes due to the additional element-wise cosine computation. Furthermore, the sinusoidal encoder tends to incur higher peak GPU memory usage for two main reasons. First, the cosine activation introduces an additional intermediate tensor during computation. Second, the linear time encoder dimension can be reduced to 1 for DyGFormer variants due to the subsequent linear projection layer as mentioned in Section 4.1.3, resulting in lower memory usage than the sinusoidal encoder.

### D.5 Beyond Pure Self-Attention Architectures

We conduct smaller-scale experiments to test if the effectiveness of the linear time encoder generalizes to model architectures beyond TGAT and DyGFormer. We experiment with a memory-based model, TGN (Rossi et al., 2020), and a random walk-based model, CAWN (Wang et al., 2021c). The results are shown in Table 9. The linear time encoder outperforms the default sinusoidal time encoder in 7 out of 8 evaluated cases. These results are consistent with our earlier findings on TGAT and the DyGFormer variants.

### D.6 Dynamic Node Classification

In this work, we focus primarily on future link prediction, as it is the most widely studied task in dynamic graph learning and benefits from a wealth of publicly available datasets. This task is also fundamental, as it evaluates a model's ability to capture the inherent temporal structure of dynamic graphs. To complement our main results, we conduct smaller-scale experiments on dynamic node classification using the two datasets provided in DyGLib (Yu et al., 2023): Wikipedia and Reddit. Following DyGLib's protocol, we first train the temporal embedding backbone using the future link prediction objective, then train a node classification head while keeping the backbone fixed. The results are presented in Table 10. On these datasets, TGAT performs better with the linear time encoder, while DyGFormer performs better with the sinusoidal time encoder,

Table 7: Training time per epoch (in seconds) across datasets and model variants. The shorter training time for each model-dataset combination is highlighted in **bold**.

| Training time / epoch (s) | Time Encoder | UCI | Wikipedia | Enron | Reddit | Last FM | USLegis |
|---|---|---|---|---|---|---|---|
| TGAT | Sinusoidal | 30.19 | 75.18 | 65.85 | 1219.34 | 734.51 | 31.38 |
| | Linear | **29.79** | **73.99** | **65.40** | **1215.03** | **732.40** | **31.18** |
| DyGFormer | Sinusoidal | 24.72 | 59.00 | 82.15 | 472.43 | 1607.48 | 40.39 |
| | Linear | **23.12** | **53.30** | **70.45** | **274.52** | **1087.59** | **35.25** |
| DyGFormer-separate | Sinusoidal | 24.87 | 59.07 | 85.51 | 298.74 | 1481.35 | 36.54 |
| | Linear | **22.76** | **58.58** | **73.08** | **282.65** | **994.33** | **32.94** |
| DyGDecoder | Sinusoidal | **24.71** | 62.83 | 85.91 | 323.38 | 1485.37 | 39.82 |
| | Linear | 24.84 | **58.80** | **64.99** | **309.53** | **1370.09** | **33.76** |

Table 8: Peak GPU memory usage (in GB) across datasets and model variants. The fewer GPU memory usage for each model-dataset combination is highlighted in **bold**.

| GPU memory usage (GB) | Time Encoder | UCI | Wikipedia | Enron | Reddit | Last FM | USLegis |
|---|---|---|---|---|---|---|---|
| TGAT | Sinusoidal | 1.930 | 1.997 | 1.972 | 2.332 | 2.728 | 1.930 |
| | Linear | **1.793** | **1.861** | **1.836** | **2.196** | **2.590** | **1.793** |
| DyGFormer | Sinusoidal | 1.024 | 1.094 | 1.581 | 1.509 | 3.180 | 1.544 |
| | Linear | **1.002** | **0.729** | **1.505** | **1.121** | **2.796** | **1.463** |
| DyGFormer-separate | Sinusoidal | 0.963 | **0.709** | 1.642 | 1.136 | 3.125 | 1.232 |
| | Linear | **0.621** | 1.006 | **1.448** | **1.087** | **2.741** | **1.037** |
| DyGDecoder | Sinusoidal | 0.646 | 1.046 | 1.671 | 1.467 | 3.141 | 1.627 |
| | Linear | **0.622** | **1.025** | **1.083** | **1.421** | **2.753** | **1.039** |

Table 9: Test AP (random NS) of TGN and CAWN with sinusoidal and linear time encoders. The better performance for each model-dataset combination is **boldfaced**.

| Test AP (random NS) | Time Encoder | UCI | Wikipedia | Enron | Reddit |
|---|---|---|---|---|---|
| TGN | Sinusoidal | $87.00_{\pm0.96}$ | $97.91_{\pm0.09}$ | $86.92_{\pm1.21}$ | $98.51_{\pm0.05}$ |
| | Linear | $\mathbf{94.53}_{\pm0.39}$ | $\mathbf{98.36}_{\pm0.05}$ | $\mathbf{90.03}_{\pm0.45}$ | $\mathbf{98.61}_{\pm0.03}$ |
| CAWN | Sinusoidal | $95.18_{\pm0.06}$ | $98.74_{\pm0.03}$ | $91.09_{\pm0.10}$ | $\mathbf{99.12}_{\pm0.01}$ |
| | Linear | $\mathbf{96.21}_{\pm0.22}$ | $\mathbf{98.99}_{\pm0.03}$ | $\mathbf{91.54}_{\pm0.28}$ | $99.10_{\pm0.01}$ |

Table 10: Test AU-ROC of TGAT and DyGFormer with sinusoidal and linear time encoders on dynamic node classification datasets. The better performance for each model-dataset combination is **boldfaced**.

| Test AU-ROC | Time Encoder | Wikipedia | Reddit |
|---|---|---|---|
| TGAT | Sinusoidal | $84.08_{\pm1.90}$ | $68.77_{\pm1.49}$ |
| | Linear | $\mathbf{86.35}_{\pm0.59}$ | $\mathbf{70.18}_{\pm1.79}$ |
| DyGFormer | Sinusoidal | $\mathbf{87.54}_{\pm1.90}$ | $\mathbf{68.76}_{\pm2.13}$ |
| | Linear | $86.33_{\pm0.97}$ | $67.51_{\pm1.02}$ |

suggesting that the two encoders are comparable for this task. A more comprehensive evaluation across additional datasets is needed to better understand their relative effectiveness in dynamic node classification, which we leave for future work.

## D.7 Attention Score Analysis

In Section 4.3.2, Figure 7 visualizes the attention scores between query and key vectors for the historical edge event sequences of source nodes in Reddit, along with the encoded time differences $t - t_q$ and $t - t_k$, respectively. We present similar visualizations for the destination nodes of Reddit in Figure 8 and five other datasets in Figures 9, 10, 11, and 12. The attention patterns observed in the continuous-time datasets align with those described in Section 4.3.2: in DyGFormer-separate, the attention scores are concentrated near

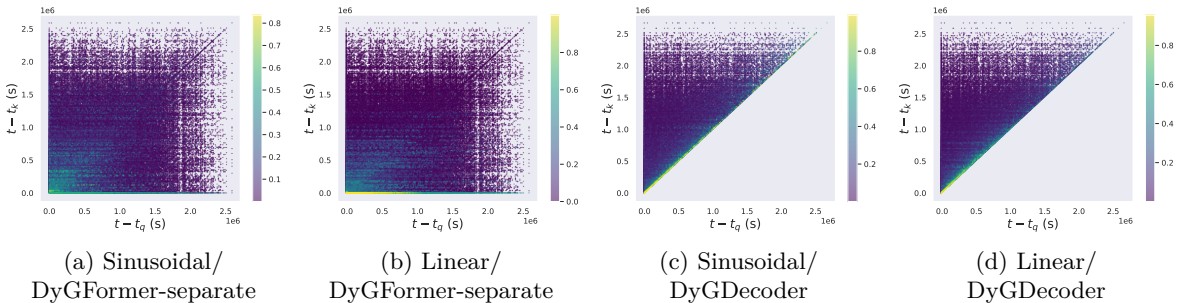

(a) Sinusoidal/
DyGFormer-separate

(b) Linear/
DyGFormer-separate

(c) Sinusoidal/
DyGDecoder

(d) Linear/
DyGDecoder

Figure 8: Attention scores (expressed in colors) of DyGFormer-separate and DyGDecoder with the sinusoidal and linear time encoders vs. $t - t_k$ vs. $t - t_q$ of the historical edge event sequences of **Reddit's destination nodes**.

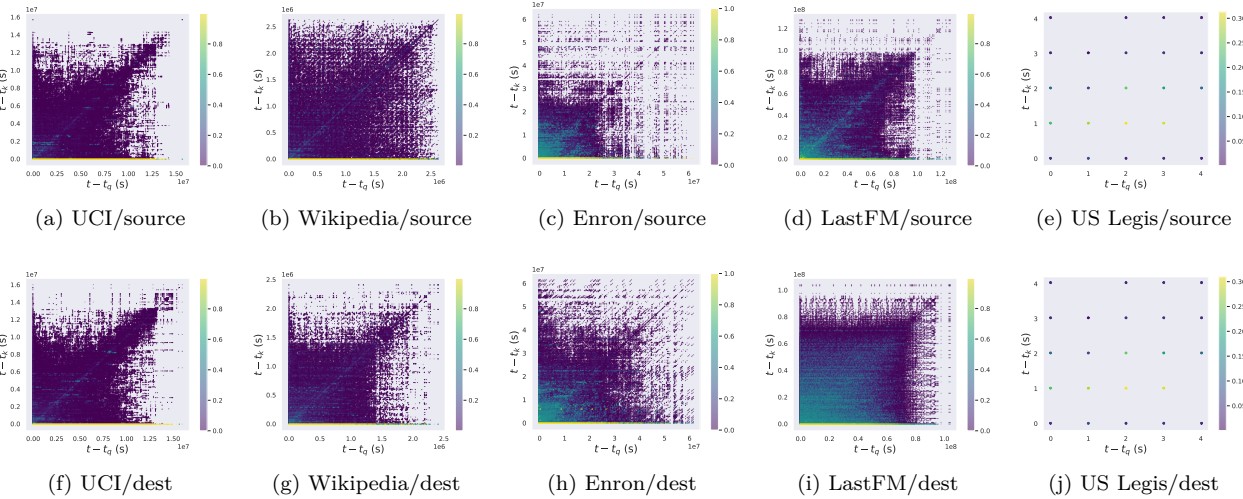

(a) UCI/source

(b) Wikipedia/source

(c) Enron/source

(d) LastFM/source

(e) US Legis/source

(f) UCI/dest

(g) Wikipedia/dest

(h) Enron/dest

(i) LastFM/dest

(j) US Legis/dest

Figure 9: Attention scores (expressed in colors) of **DyGFormer-separate** with the **sinusoidal time encoder** vs. $t - t_k$ vs. $t - t_q$ of the historical edge event sequences of nodes in UCI, Wikipedia, Enron, LastFM, and US Legis. The upper and bottom rows are the results of the source and destination nodes.

$t - t_k = 0$, and in DyGDecoder, they focus around $t - t_k = t - t_q$. However, attention patterns can vary across datasets. For instance, in DyGFormer-separate, the attention scores for UCI are tightly concentrated, while for LastFM, the scores are more spread out across the $(t - t_q)$-$(t - t_k)$ plane, reflecting distinct characteristics of the datasets.

Being a discrete-time dataset, US Legis has a limited number of distinct time differences, as seen in Figures 9e, 9j, 10e, 10j, 11e, 11j, 12e, and 12j. In this case, we average all attention scores corresponding to the same $(t - t_k, t - t_q)$ pair and visualize the result. Interestingly, in DyGFormer-separate, the highest attention scores are not located at $t - t_k = 0$, but at $t - t_k = 1$. This is because the key vectors corresponding to the most recent events before the target time $t$ are located at $t - 1$, and these key vectors have the greatest influence on the node's state at time $t$. In contrast, the key vectors corresponding to the target edges lack informative edge features, as the interaction has not yet occurred. Consequently, these key vectors are less important for computing a node's temporal representation. This should also hold for continuous-time datasets, where the key vectors encoding $t - \epsilon$ should have the highest attention scores, with $\epsilon$ being a small positive quantity. However, this effect is not easily discernible in the plots due to the continuous nature of the time differences being visualized.

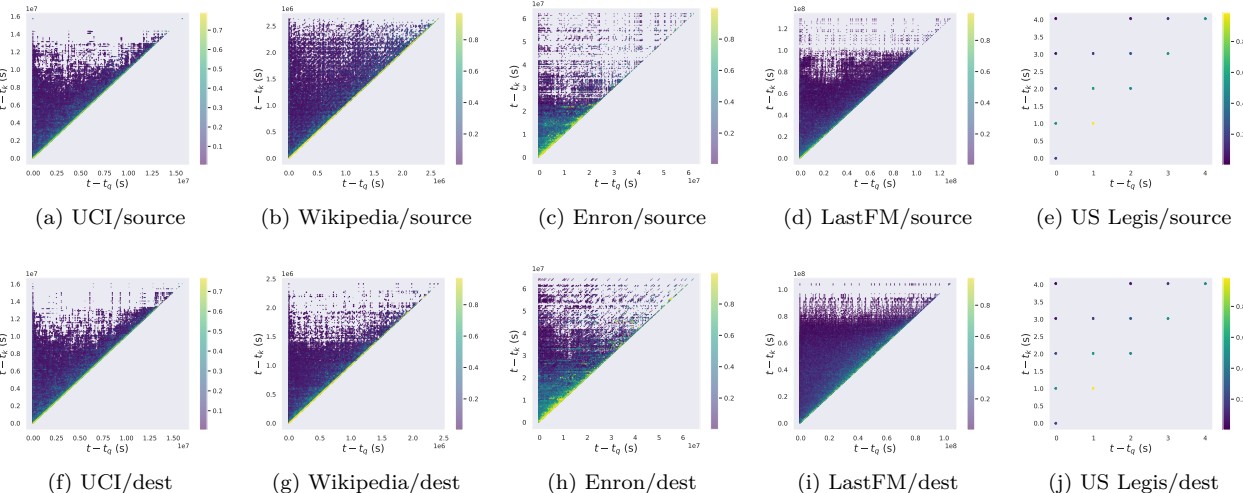

Figure 10: Attention scores (expressed in colors) of **DyGDecoder** with the **sinusoidal time encoder** vs. $t - t_k$ vs. $t - t_q$ of the historical edge event sequences of nodes in UCI, Wikipedia, Enron, LastFM, and US Legis. The upper and bottom rows are the results of the source and destination nodes.

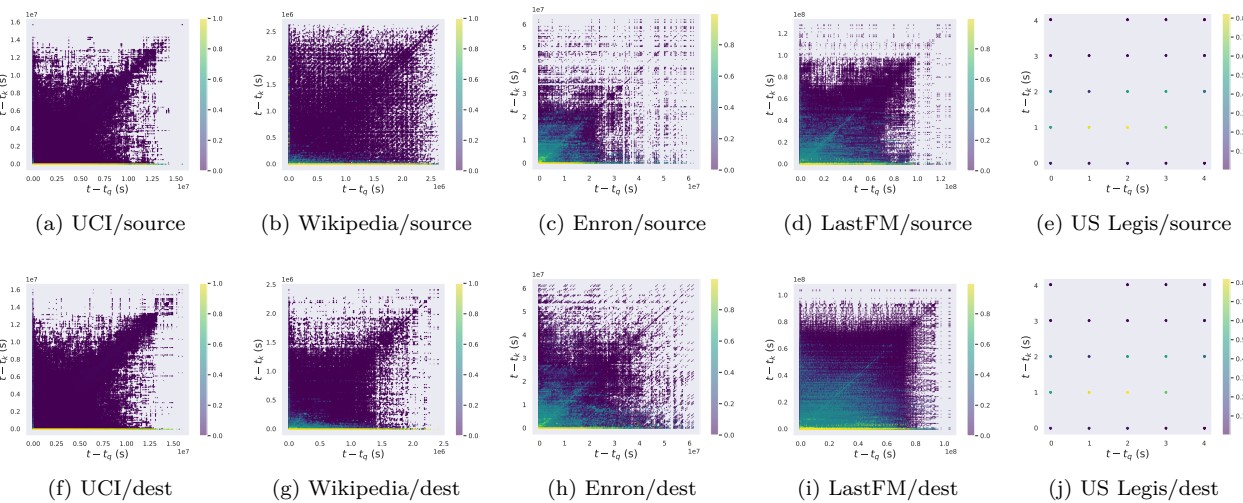

Figure 11: Attention scores (expressed in colors) of **DyGFormer-separate** with the **linear time encoder** vs. $t - t_k$ vs. $t - t_q$ of the historical edge event sequences of nodes in UCI, Wikipedia, Enron, LastFM, and US Legis. The upper and bottom rows are the results of the source and destination nodes.

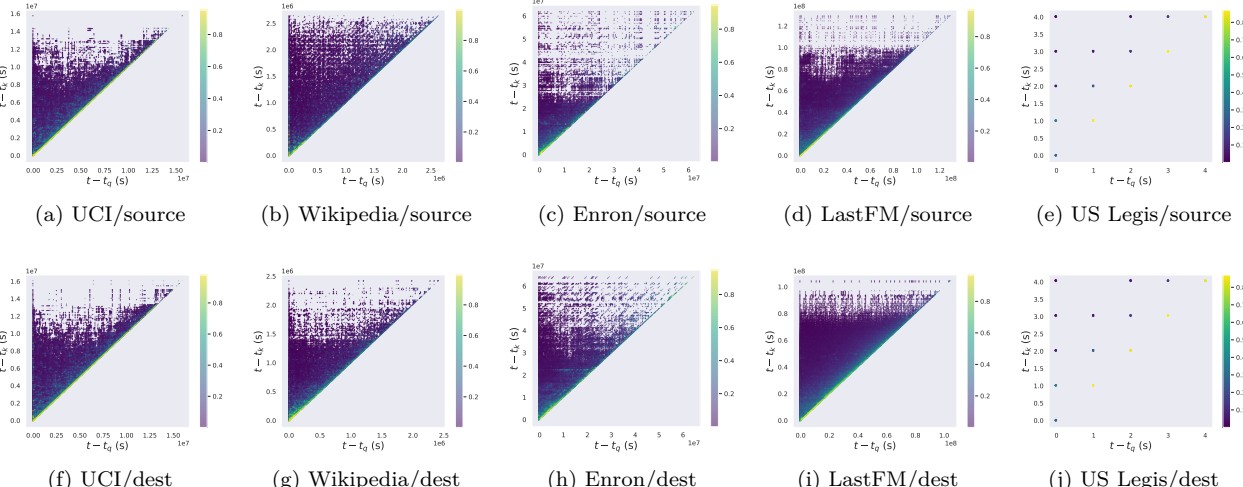

Figure 12: Attention scores (expressed in colors) of **DyGDecoder** with the **linear time encoder** vs. $t - t_k$ vs. $t - t_q$ of the historical edge event sequences of nodes in UCI, Wikipedia, Enron, LastFM, and US Legis. The upper and bottom rows are the results of the source and destination nodes.

