# OpenReview forum: "Between Linear and Sinusoidal: Rethinking the Time Encoder in Dynamic Graph Learning"
_TMLR — Accepted by TMLR_

### Review · Reviewer_Comy · 2025-05-08

**Summary Of Contributions:**

The paper challenges the de‑facto use of high‑dimensional sinusoidal time encoders in dynamic‑graph Transformers. It proposes a linear time encoder that feeds the raw time gap Δt through a learnable linear projection

**Audience:**

Yes

**Claims And Evidence:**

Yes

**Requested Changes:**

- Add one memory‑based and one walk‑based architecture to the test bed; even a small‑scale run would show the idea generalises.
- Add a non‑link task so we know the results aren’t benchmark‑specific.
- Provide per‑epoch time and peak GPU usage.
- In Tables 1–2, visually highlight the win counts to let readers grasp the advantage instantly.

**Strengths And Weaknesses:**

## Strengths
- The paper is well-written and easy to follow. The paper challenges a small but influential design choice that everyone inherits without much thought.
- Six datasets, two transformer‑style backbones, twenty‑four runs… that’s a lot of GPU time spent on a simple idea.
- Practical impact - The attention heatmaps and the ablation curves make it clear where the gains come from.

## Weaknesses
- We only see results on TGAT and DyGFormer. I kept wondering how a memory‑centric GNN would react?
- The authors report parameter counts but not wall‑clock time.
- Narrow task scope – Future link prediction is nice, but node classification, temporal motifs, or cold‑start tests could paint a very different picture.

---

> ### Author Response · Authors · 2025-05-17
> **Response to Requested Change**
>
> Hi Reviewer Comy,
>
> Thank you for your constructive feedback. We appreciate the opportunity to address your concerns. Below, we respond to your requested changes, which also reflect the key weaknesses you identified.
>
> **Response to Requested Change 1:**\
> Thank you for the suggestion. In response, we conduct additional experiments with TGN [1] (memory-based) and CAWN [2] (walk-based) on 4 continuous-time dynamic graph datasets and present the results below. According to the results, the linear time encoder outperforms the default sinusoidal time encoder in 7 out of 8 cases. The results are consistent with our findings from TGAT and DyGFormer variants. We have included this table in the appendix of our revised paper.
>
> | Test AP (random NS) | Time Encoder | UCI | Wikipedia | Enron | Reddit |
> |:---------------|:------------:|:----------------:|:----------------:|:----------------:|:----------------:|
> | TGN | Sinusoidal | 87.00 ± 0.96 | 97.91 ± 0.09 | 86.92 ± 1.21 | 98.51 ± 0.05 |
> |  | Linear | **94.53** ± 0.39 | **98.36** ± 0.05 | **90.03** ± 0.45 | **98.61** ± 0.03 |
> | CAWN | Sinusoidal | 95.18 ± 0.06 | 98.74 ± 0.03 | 91.09 ± 0.10 | **99.12** ± 0.01 |
> |  | Linear | **96.21** ± 0.22 | **98.99** ± 0.03 | **91.54** ± 0.28 | 99.10 ± 0.01 |
>
>
>
> **Response to Requested Change 2:**\
> In this work, we focus on future link prediction since it is the most widely studied task in dynamic graph learning with abundant publicly available datasets. It is also fundamental, as it tests the model’s understanding of the dynamic graph’s inherent structure. That being said, we agree that evaluating on other tasks would provide additional insights. We conduct node classification experiments with TGAT and DyGFormer with the sinusoidal and linear time encoder on Wikipedia. The two time encoders are comparable on Wikipedia where TGAT is better with the linear encoder and DyGFormer is better with the sinusoidal encoder. We are still running additional node classification experiments and will update new results during the rebuttal period.
>
> | Test AU-ROC | Time Encoder | Wikipedia |
> |:---------------|:------------:|:----------------:|
> | TGAT | Sinusoidal | 84.08 ± 1.90 |
> |  | Linear | **86.35** ± 0.59 |
> | DyGFormer | Sinusoidal | **87.54** ± 1.90 |
> |  | Linear | 86.33 ± 0.97 |
>
>
> [1] Rossi et al. Temporal Graph Networks for Deep Learning on Dynamic Graphs. ICML 2020 Workshop on Graph Representation Learning. \
> [2] Wang et al. Inductive Representation Learning in Temporal Networks via Causal Anonymous Walks. ICLR 2021.

---

> > ### Author Response · Authors · 2025-05-17
> > **Response to Requested Change (Cont'd)**
> >
> > **Response to Requested Change 3:**\
> > As requested, we report the per-epoch training time and peak GPU memory usage of sinusoidal and linear time encoders on each model-dataset combination below. The runtime when using the sinusoidal time encoder is usually longer since there is an extra element-wise cosine computation compared to the linear time encoder. The sinusoidal time encoder also leads to higher peak GPU memory usage for two reasons. First, the cosine activation introduces an extra intermediate tensor. Second, the linear time encoder dimension can be reduced to 1 for DyGFormer variants due to the linear projection layer as mentioned in Section 4.1.3, resulting in lower memory usage than the sinusoidal encoder. The detailed runtime and memory statistics are now included in the appendix of the revised paper to inform readers about the computational cost.
> >
> >
> > | Training time / epoch (s) | Time Encoder | UCI   | Wikipedia | Enron | Reddit   | Last FM  | USLegis |
> > |---------------------------|:--------------:|:-------:|:-----------:|:-------:|:----------:|:----------:|:---------:|
> > | **TGAT**                  | Sinusoidal   | 30.19 | 75.18     | 65.85 | 1219.34  | 734.51   | 31.38   |
> > |                           | Linear       | **29.79** | **73.99**     | **65.40** | **1215.03** | **732.40**  | **31.18**  |
> > | **DyGFormer**             | Sinusoidal   | 24.72 | 59.00     | 82.15 | 472.43   | 1607.48  | 40.39   |
> > |                           | Linear       | **23.12** | **53.30**     | **70.45** | **274.52**  | **1087.59** | **35.25**  |
> > | **DyGFormer-separate**    | Sinusoidal   | 24.87 | 59.07     | 85.51 | 298.74   | 1481.35  | 36.54   |
> > |                           | Linear       | **22.76** | **58.58**     | **73.08** | **282.65**  | **994.33**  | **32.94**  |
> > | **DyGDecoder**            | Sinusoidal   | **24.71** | 62.83     | 85.91 | 323.38   | 1485.37  | 39.82   |
> > |                           | Linear       | 24.84 | **58.80**     | **64.99** | **309.53**  | **1370.09** | **33.76**  |
> >
> >
> > | GPU memory usage (GB)     | Time Encoder | UCI   | Wikipedia | Enron | Reddit | Last FM | USLegis |
> > |---------------------------|:--------------:|:-------:|:-----------:|:-------:|:--------:|:---------:|:---------:|
> > | **TGAT**                  | Sinusoidal   | 1.930 | 1.997     | 1.972 | 2.332  | 2.728   | 1.930   |
> > |                           | Linear       | **1.793** | **1.861**     | **1.836** | **2.196**  | **2.590**   | **1.793**   |
> > | **DyGFormer**             | Sinusoidal   | 1.024 | 1.094     | 1.581 | 1.509  | 3.180   | 1.544   |
> > |                           | Linear       | **1.002** | **0.729**     | **1.505** | **1.121**  | **2.796**   | **1.463**   |
> > | **DyGFormer-separate**    | Sinusoidal   | 0.963 | **0.709**     | 1.642 | 1.136  | 3.125   | 1.232   |
> > |                           | Linear       | **0.621** | 1.006     | **1.448** | **1.087**  | **2.741**   | **1.037**   |
> > | **DyGDecoder**            | Sinusoidal   | 0.646 | 1.046     | 1.671 | 1.467  | 3.141   | 1.627   |
> > |                           | Linear       | **0.622** | **1.025**     | **1.083** | **1.421**  | **2.753**   | **1.039**   |
> >
> >
> >
> >
> >
> >
> >
> >
> > **Response to Requested Change 4:**\
> > Thank you for this helpful suggestion. We have updated Tables 1 and 2 to boldface the highest win counts corresponding to each model variant. We hope this improves readability and allows readers to quickly grasp the comparative advantage of the linear time encoder.
> >
> >
> > We hope that you will find our responses satisfactory. The modified parts in the paper are highlighted and we would be happy to answer any further questions.
> >
> > Sincerely,\
> > Authors

---

> > > ### Author Response · Authors · 2025-05-23
> > >
> > > **Response to Requested Change 2 (Cont’d):**\
> > > We have completed additional node classification experiments on the Reddit dataset, the other node classification dataset provided in DyGLib alongside Wikipedia. The full results are presented in the table below. Consistent with the findings on Wikipedia, TGAT performs better with the linear time encoder, while DyGFormer performs better with the sinusoidal encoder. These results suggest that the two time encoders are currently comparable for dynamic node classification, with performance depending on the model architecture. While our current evaluation covers all node classification datasets available in DyGLib, a broader assessment on more datasets would be valuable for understanding the relative effectiveness of the time encoders, and we consider this a direction for future work. We have included this updated table in the appendix of the latest revision. Although the two time encoders perform comparably on the two node classification datasets, the linear time encoder generally achieves stronger results and better parameter/computational efficiency in future link prediction, highlighting its value in the most widely studied task.
> > >
> > >
> > > | Test AU-ROC | Time Encoder | Wikipedia | Reddit |
> > > |:---------------|:------------:|:----------------:|:----------------:|
> > > | TGAT | Sinusoidal | 84.08 ± 1.90 | 68.77 ± 1.49 |
> > > |  | Linear | **86.35** ± 0.59 | **70.18** ± 1.79 |
> > > | DyGFormer | Sinusoidal | **87.54** ± 1.90 | **68.76** ± 2.13 |
> > > |  | Linear | 86.33 ± 0.97 | 67.51 ± 1.02 |
> > >
> > > We hope that our response to the overall requested changes, including the addition of a new task, the evaluation of non-attention-centric model architectures, and the reporting of computational resources, adequately addresses your concerns.
> > >
> > > Sincerely,\
> > > Authors

---

### Review · Reviewer_v7gf · 2025-05-09

**Summary Of Contributions:**

This paper investigates the use of a linear time encoder as an alternative to the more commonly used sinusoidal time encoder in attention-based dynamic graph learning models, specifically TGAT and DyGFormer. The authors argue that linear time encoders can prevent temporal information loss associated with sinusoidal functions and potentially reduce the need for high-dimensional time encodings. The paper provides a theoretical justification showing that self-attention mechanisms can effectively learn to compute time spans from linear time encodings. This is further supported by experiments on a synthetic dataset. Extensive experiments on six real-world dynamic graph datasets for future link prediction demonstrate that the linear time encoder generally outperforms sinusoidal encoders in terms of average precision and can lead to significant reductions in model parameters with minimal performance degradation.

**Audience:**

Yes

**Claims And Evidence:**

Yes

**Requested Changes:**

In addition the the mentioned weaknesses:

1.  Proposition 3.1 is interesting, showing existence. However, the construction in Appendix B seems specific. Could the authors elaborate if common initialization schemes or training dynamics naturally lead to such solutions, or if specific architectural choices beyond the linear encoder itself facilitate this? The synthetic task (Section 3.2) shows it *can* be learned, but further insight into *how* readily it's learned in complex models would be valuable.
2.  The paper sets the linear time encoding dimension to 100 for TGAT but to 1 for DyGFormer variants (Section 4.1.3), explaining that the latter is followed by a linear projection. While the dimensionality reduction experiments (Section 4.3.1) are insightful, a brief discussion on the sensitivity to the *initial* dimension choice for linear encoders, especially if the subsequent projection was not present, might be useful. For example, how critical is $d_T=1$ for DyGFormer if the channel dimension $d_{ch}$ is also small?
3.  The paper introduces "DyGDecoder" as a variant computing representations "separately and in an autoregressive manner" and prepending a "beginning of sequence" embedding (Section 3.3). While Figure 6 shows attention patterns, a more explicit (even if brief) architectural diagram or description of how the autoregressive computation and the BOS token are integrated specifically for the link prediction task would enhance clarity. How does the "target edge event" fit into this autoregressive formulation when its features are zeroed out (as mentioned for DyGFormer in Section 2.2)?
6.  The paper notes that US Legis is a "discrete-time dynamic graph with only 12 evenly spaced time steps" (Section 4.2.3). Given this, are time differences always integer multiples of some base unit? If so, this is quite different from the continuous-time nature of other datasets. Clarifying this might further explain why sinusoidal encoders might be sufficient or even preferable here, as the "discretization" inherent in sinusoidal functions over a limited set of inputs might align well with the data's temporal structure.
7.  The caption for Figure 5 refers to "varying time channel dimension $d_{tc}$". However, the plots themselves and the text (Section 4.3.1) seem to vary $d_{ch}$ (channel embedding dimension) or effectively $d_T$ (time encoding dimension, which is 1 and then projected to $d_{ch}$ for the linear case in DyGFormer). It would be good to ensure consistent terminology or clarify if $d_{tc}$ is a distinct hyperparameter being varied. My understanding from Section 2.2 and 4.1.3 is that $d_{ch}$ is the key dimension here for DyGFormer.

**Strengths And Weaknesses:**

Strengths:
1.  The paper challenges a widely adopted component (sinusoidal time encoder) in dynamic graph learning and proposes a simpler, yet effective, alternative. The findings have direct practical implications for designing more efficient dynamic graph learning architectures.
2.  The authors provide a proof (Proposition 3.1) that self-attention can indeed learn to factor time spans from linear time encodings, which is a key motivation for using sinusoidal encoders. This theoretical insight is valuable.
3.  The paper presents a comprehensive experimental evaluation across six diverse datasets and two different negative sampling strategies. The inclusion of a synthetic task to validate the learnability of time spans from linear encodings (Section 3.2) strengthens the claims.

Weaknesses:
1.  While TGAT and DyGFormer are representative, the study is confined to these two attention-based architectures. It would be interesting to see if the conclusions generalize to other types of dynamic graph models, including those not heavily reliant on self-attention or those using different mechanisms for incorporating time.
2.  While the linear encoder shows overall strong performance, there are instances where it does not outperform or even underperforms sinusoidal encoders, for example, on the US Legis dataset under historical negative sampling for most model variants (Table 2). The paper offers a plausible explanation regarding the discrete nature of US Legis, but further investigation into why linear encoders might struggle in certain data regimes could be beneficial.
3.  The paper mentions that standardized time differences are used for "numerical stability and optimization efficiency purposes". While the sinusoidal-scale encoder is introduced to isolate the impact of scaling, the interaction between standardization and the performance of linear encoders, especially in scenarios with significant distribution shifts in time differences (as noted for UCI in Section 4.2.2), could be a point of deeper discussion. It's not entirely clear if the performance fluctuations are solely due to the scaling strategy or an inherent property of linear encoding in such scenarios.

---

> ### Author Response · Authors · 2025-05-17
> **Response to Weaknesses**
>
> Hi Reviewer v7gf,
>
> Thank you for reading our paper and writing a detailed, thoughtful review. We appreciate the opportunity to respond to your comments and suggestions. Below, we address the listed weaknesses first, followed by the requested changes.
>
> **Response to Weakness 1:**\
> Our study primarily focuses on TGAT and DyGFormer variants due to the prevalence of self-attention mechanisms in modern neural architectures and the fact that the commonly used sinusoidal time encoder in dynamic graph learning is motivated by self-attention [1]. That said, we agree that it would be interesting to see if our results generalize to other types of architectures. Therefore, we conduct additional experiments on a memory-based model, TGN [2], and a random walk-based model, CAWN [3], on 4 continuous-time dynamic graph datasets. The results are presented in the table below. According to the results, the linear time encoder outperforms the default sinusoidal encoder in 7 out of 8 cases. The results align with our observation of TGAT and DyGFormer variants and show that our conclusion generalizes to other architectures. We have added this table to the appendix of the revised paper.
>
> | Test AP (random NS) | Time Encoder | UCI | Wikipedia | Enron | Reddit |
> |:---------------|:------------:|:----------------:|:----------------:|:----------------:|:----------------:|
> | TGN | Sinusoidal | 87.00 ± 0.96 | 97.91 ± 0.09 | 86.92 ± 1.21 | 98.51 ± 0.05 |
> |  | Linear | **94.53** ± 0.39 | **98.36** ± 0.05 | **90.03** ± 0.45 | **98.61** ± 0.03 |
> | CAWN | Sinusoidal | 95.18 ± 0.06 | 98.74 ± 0.03 | 91.09 ± 0.10 | **99.12** ± 0.01 |
> |  | Linear | **96.21** ± 0.22 | **98.99** ± 0.03 | **91.54** ± 0.28 | 99.10 ± 0.01 |
>
>
>
>
> **Response to Weakness 2:**\
> We acknowledge that there are cases where the linear time encoder does not consistently outperform the sinusoidal encoder. Based on our observation, those cases are due to: (i) scaling under distribution shift (UCI), (ii) the dataset being relatively insensitive to temporal features (Reddit), and (iii) the dataset having coarse-grained temporal features (US Legis). However, even in cases where the linear encoder underperforms, its performance is typically within one standard deviation of the sinusoidal baseline. In contrast, in the scenarios where the linear encoder excels, the improvement is often substantial. We discuss the UCI case further in “Response to Weakness 3” and address the US Legis case in “Response to Requested Change 4”.
>
>
> **Response to Weakness 3:**\
> Thank you for asking about the effects of scaling and the linearity of the time encoder under distribution shift (UCI). There are two interventions as we change from the sinusoidal encoder to sinusoidal-scale, and to the linear time encoder. The first one is scaling (sinusoidal &rarr; sinusoidal-scale) and the second one is linearity (sinusoidal-scale &rarr; linear). On UCI, the performances of all model variants decrease as scaling is applied to sinusoidal encoders, with the greatest performance drop being 11.25 AP scores. This negative effect is not seen in other datasets with no significant distribution shifts. When the linearity is applied subsequently (sinusoidal-scale &rarr; linear), the performances of all model variants bounce back on UCI. This positive effect is consistent with most other datasets. Therefore, we can attribute most of the negative performance fluctuations under distribution shift to scaling. In fact, when scaling is already applied, the linearity of the time encoder helps under distribution shift. This echoes the findings in the literature on relative positional encoding of language models [4] which discovered that the linear function of positional differences enables strong length generalization. We have revised the second paragraph of Section 4.2.2. to emphasize the negative effect of scaling under distribution shift.
>
> [1] da Xu et al. Inductive representation learning on temporal graphs. ICLR 2020.\
> [2] Rossi et al. Temporal Graph Networks for Deep Learning on Dynamic Graphs. ICML 2020 Workshop on Graph Representation Learning. \
> [3] Wang et al. Inductive Representation Learning in Temporal Networks via Causal Anonymous Walks. ICLR 2021.\
> [4] Press et al. Train Short, Test Long: Attention with Linear Biases Enables Input Length Extrapolation. ICLR 2022.

---

> > ### Author Response · Authors · 2025-05-17
> > **Response to Requested Changes**
> >
> > **Response to Requested Change 1:**\
> > Thank you for reading through the proposition proof. The proposition shows existence by only specifying two dimensions of parameters and the synthetic experiments complement the proposition by showing that with common initialization (Kaiming) and training (Adam optimizer), transformers can factor the time spans to the attention scores from linear time encodings. Regarding how “readily” the time encoder and transformers are learned when the model is more complex and with more parameters, we examine the early phase of the learning curves in the main experiments on dynamic graph datasets. In specific, we report the validation AP scores with linear and sinusoidal time encoders after only 1 epoch of training. We observe that the linear encoder outperforms 20 out of 24 cases, implying that it is more readily learned such that it achieves better performances with only a single pass over each dataset.
> >
> >
> > | Validation AP (random NS)       | Time Encoder | UCI   | Wikipedia | Enron | Reddit | Last FM | USLegis |
> > |--------------------------|:--------------:|:-------:|:-----------:|:-------:|:--------:|:---------:|:---------:|
> > | TGAT                     | Sinusoidal   | 83.12 | 93.16     | 64.00    | **95.98**  | 65.21   | 50.40    |
> > |                          | Linear       | **90.59** | **94.27** | **73.53** | 94.82 | **74.07** | **63.72** |
> > | DyGFormer                | Sinusoidal   | **93.49** | 98.97     | **85.84** | 98.88  | **90.23**   | 61.75   |
> > |                          | Linear       | 90.75 | **99.05**  | 80.02 | **98.95**  | 90.16   | **66.72**   |
> > | DyGFormer-separate       | Sinusoidal   | 93.76 | 99.01     | 77.78 | 98.92  | 90.55   | 60.64   |
> > |                          | Linear       | **93.90** | **99.27** | **85.22** | **99.03** | **90.63** | **63.84** |
> > | DyGDecoder               | Sinusoidal   | 94.01 | 99.17     | 85.66 | 98.96  | 91.74   | 58.26   |
> > |                          | Linear       | **95.51** | **99.26**  | **86.21** | **99.01**  | **92.09**   | **59.71**   |
> >
> >
> >
> >
> > **Response to Requested Change 2:**\
> > In DyGFormer, a projection layer is applied to ensure that the four input channels (neighbor node features, edge features, temporal features, and neighbor co-occurrence features) share a consistent dimension before entering the self-attention block. For the linear time encoder, the initial dimensionality is inconsequential as long as the projection layer is used since there are no non-linearities in between. The composition of a linear encoder followed by a projection is functionally equivalent to a linear encoder with an output dimension equal to the channel size. If the projection were removed, the initial dimension would become critical and would need to match the channel size. However, removing such layers would diverge from the standard model architecture and detract from our core research questions, so we chose not to explore that direction.
> >
> >
> > **Response to Requested Change 3:**\
> > Thank you for the suggestion. We have added Figure 4 to the revised paper to illustrate the similarities and differences between the DyGFormer variants and modified the text to provide clarity. The learnable “beginning of sequence” embedding is prepended to the input sequence feature matrices of the source and destination nodes. The autoregressive computation simply uses a transformer with causal masking as described in Section 3.2.2. The features of the target edge events are placed at the last vectors of the input feature sequence matrices. The corresponding edge attributes are zeroed out since the target edge has not yet occurred. This does not affect the autoregressive computation of DyGDecoder since the autoregressive computation is independent of the input feature preparation. Please refer to Appendix A.2 for further details on the input feature preparation of DyGFormer.

---

> > > ### Author Response · Authors · 2025-05-17
> > > **Response to Requested Changes (Cont'd)**
> > >
> > > **Response to Requested Change 4:**\
> > > Yes, your understanding is correct. In US Legis, the time differences are always integer multiples of the base unit of congressional sessions. There are only 12 possible time differences: {0, 1, …, 11}. Since the time granularity of the dataset is already coarse-grained, it is easy for the sinusoidal time encoder to preserve temporal information. For example, a one-dimensional sinusoidal time encoder such as $\cos(\frac{\pi}{11} \Delta t)$ can encode the limited set of possible time differences into the range of [-1, 1] as a one-to-one mapping without collision (see the table below).
> > >
> > > | $\Delta t$ | 0 | 1 | 2 | 3 | 4 | 5 | 6 | 7 | 8 | 9 | 10 | 11 |
> > > |------------|:----:|:----:|:----:|:----:|:----:|:----:|:----:|:----:|:----:|:----:|:----:|:----:|
> > > | $ \cos\Bigl(\tfrac{\pi}{11}\Delta t\Bigr)$ | 1 | 0.959 | 0.841 | 0.655 | 0.415 | 0.142 | -0.142 | -0.415 | -0.655 | -0.841 | -0.959 | -1 |
> > >
> > > However, if we apply the same trick to other continuous-time datasets with the base unit in seconds and the maximum possible time difference $T$ in the scale of $10^6$ seconds using a time encoder $\cos(\frac{\pi}{T} \Delta t)$, the continuous time differences will be mapped to the range [-1, 1] with potential collisions due to machine precision issues. Using $T=10^6$ and Float-32 resolution as an example, $\cos(\frac{\pi}{T} \cdot 0)$ and $\cos(\frac{\pi}{T} \cdot 85)$ will both result in 1. Therefore, this trick is not applicable and a higher dimension of sinusoidal time encoder is needed.
> > >
> > > We hope the examples above show why US Legis is simpler than other datasets and why the sinusoidal time encoder is sufficient for it. That said, linear encoders perform comparably on US Legis and win half of the time when considering both random and historical negative sampling evaluation. Therefore, we refrain from claiming a clear preference for one over the other on US Legis. We have revised the last paragraph in Section 4.2.3 to add further explanations.
> > >
> > >
> > > **Response to Requested Change 5:**\
> > > Thank you for pointing this out. In the original DyGFormer before this auxiliary experiment, all channels have the same number of dimensions $d_{ch}$. In this auxiliary experiment, we varied specifically the temporal feature channel dimension but fixed the dimensions of the node, edge, and neighbor co-occurrence feature channels. Therefore, we use a new notation $d_{tc}$ to denote the temporal feature channel dimension separately since it may be different from the dimensions of other channels. We have revised Section 4.3.1 in the paper to introduce the notation difference and we hope this brings clarity to the readers.
> > >
> > > We hope our responses have addressed your concerns. We have highlighted the revised parts in the paper and remain available to provide further clarification.
> > >
> > > Sincerely,\
> > > Authors

---

### Review · Reviewer_nqHa · 2025-05-09

**Summary Of Contributions:**

This paper proposes to use a linear time encoder instead of a sinusoidal time encoder for dynamic graph learning. The author demonstrates that this new approach can outperform the baseline in most cases while also saving memory through many experiments.

**Audience:**

Yes

**Claims And Evidence:**

Yes

**Requested Changes:**

1. In Sections 2.3 and 3.1, the variable w is used as a learnable weight, but it is only explained in Section 3.1. For clarity, this explanation should be included in Section 2.3. Additionally, in the first two formulas of Section 2.3, the use of d should be replaced with d_T.
2. Please double-check the value reported in Table 1 for DyGFormer-separate with sinusoidal-scale on the UCI dataset: 87.77 +/- 15.75. The standard deviation of 15.75 seems unusually high, could this be a typo?
3. The author provides many figures of attention scores from DyGFormer-separate and DyGDecoder in both the main text and the appendix, which help illustrate the model’s behavior. In Section 4.3.2, the paper discusses the role of pooling strategies and highlights the importance of edge event timing in dynamic graphs. While these insights are valid, they should serve as the foundation of the paper rather than the main focus. The central contribution of the paper is the imprvoement of the proposed linear encoder, and I expect to see visual comparisons between linear encoders and sinusoidal encoders.
4. Minor point: It would be better to replace the e-commerce platform example in the introduction. When I first read the paper, this example caught my attention and led me to expect that the model would be evaluated on e-commerce-related data. However, none of the datasets used in the experiments are related to e-commerce, which ultimately felt disappointing. While the example is engaging, I recommend choosing an example that aligns more closely with the datasets used in the study.

**Strengths And Weaknesses:**

Strengths:
1. The proposed idea is intuitive, and both the formulation and theoretical justification are clearly presented.
2. The experimental results show consistent performance improvements. Although this method does not always achieve the highest score on all datasets, the performance remains within one standard deviation of the SoTA, which is acceptable.
3. The authors evaluate their approach on both DyGFormer and DyGDecoder models, demonstrating the generalizability of the new encoder.

Weakness:
See requested changes

---

> ### Author Response · Authors · 2025-05-17
>
> Hi Reviewer nqHa,
>
> Thank you for reviewing our paper and providing valuable feedback. We appreciate the chance to address your requested changes below.
>
> **Response to Requested Changes 1:**\
> Thank you for pointing this out. We have updated the notation in Section 2.3 to replace $d$ with $d_T$ in the first two equations, as suggested. While we originally mentioned that the time encoder parameters are learnable at the end of Section 2.3, we agree that an earlier clarification improves clarity. Accordingly, we have added a sentence introducing the learnable parameters when we first present the sinusoidal time encoder.
>
> **Response to Requested Changes 2:**\
> We have re-verified the reported value in Table 1, and the standard deviation of 15.75 is correct. The sinusoidal-scale encoder exhibits instability on the UCI dataset, with one run performing significantly worse than the others. Below, we provide the results from five independent runs of DyGFormer-separate using the sinusoidal-scale encoder:
>
> | Seed | 1    | 2    | 3    | 4    | 5    |
> |:------:|:------:|:------:|:------:|:------:|:------:|
> | Test AP (Random) | 59.69 | 92.75 | 95.01 | 95.91 | 95.51 |
>
>
>
> **Response to Requested Changes 3:**\
> We believe that showing and discussing the attention patterns of linear time encoders with respect to architectural choice and edge event timing is also a focus of the paper since our claim is that linear time encoders can enable transformers to capture temporal relationships. Just like we showed the attention scores of the synthetic experiments in Figure 3, we visualized the attention scores on realistic datasets to provide insights. That being said, we agree that it would be interesting to compare attention patterns between sinusoidal and linear time encoders. Therefore, we replaced the figure with a comparison between the two encoders on Reddit’s source nodes and added a paragraph of discussion in the revised paper. We put the remaining visualizations in the appendix for the reader’s reference.
>
>
> **Response to Requested Changes 4:**\
> Thank you for the suggestion. In the revised paper, we have replaced the e-commerce example with a person-to-person communication network example, which aligns with the UCI and Enron datasets we used for experiments.
>
> We hope these revisions address your concerns effectively. Please feel free to reach out with any further questions or suggestions.
>
> Sincerely,\
> Authors

---

> > ### Comment · Reviewer_nqHa · 2025-06-02
> >
> > Thank you for your response. I am satisfied with your clarification.

---

### Decision · Action_Editor_b87L · 2025-06-15

**Recommendation:** Accept as is

**Audience:**

Yes

**Audience Explanation:**

The paper challenges the standard adoption of sinusoidal time encodings for dynamic graphs and presents evidence that a linear approach offers advantages. This will be of interest to researchers working in the field of learning on graphs.

**Claims And Evidence:**

Yes

**Claims Explanation:**

The  paper makes three major claims in terms of contribution. The claims are supported by theoretical and empirical evidence.